# Can Agents Generalize to the Open World?
# Unveiling the Fragility of Static Training in Tool Use

**Weiming Wu** [* 1]  **Song-Lin Lv** [* 1 2]  **Rui Zhu** [1]  **Zi-Jian Cheng** [1 2]  **Lan-Zhe Guo** [1 2]

## Abstract

While Large Language Model (LLM) agents demonstrate proficiency in static benchmarks, their deployment in real-world scenarios is hindered by the dynamic nature of user queries, tool sets, and interaction dynamics. To address this generalization gap, we formalize **OpenAgent** (Tool-Use Agent in Open-World), a problem setting characterized by distributional shifts across query, action, observation, and domain dimensions. To systematically diagnose its impact, we construct a controlled sandbox environment where we define fine-grained environmental shifts across a four-tier hierarchy, *Perception*, *Interaction*, *Reasoning*, and *Internalization*, and conduct a comprehensive series of experiments. Our analysis yields a series of key insights, demonstrating that agents trained via both Supervised Fine-Tuning(SFT) and Reinforcement Learning suffer from varying degrees of performance degradation when confronting open environmental shifts. Building on these insights, we propose Perturbation-Augmented Fine-Tuning, a disturbance-based intervention strategy for SFT that lays the foundation for enhancing agent robustness and utility in realistic environments. Our code will be released at: https://github.com/LAMDA-NeSy/OpenAgent.

## 1. Introduction

The integration of Tool Learning and the Model Context Protocol (MCP) (Anthropic, 2024) has catalyzed a paradigm shift in Large Language Model (LLM) agents, enabling them to navigate external environments to solve complex, multi-step tasks (Guo et al., 2024; Wang et al., 2024b; Qu et al., 2025). Optimized via Supervised Fine-Tuning (SFT) and Reinforcement Learning (RL), recent open-source models (Hsieh et al., 2023; Qu et al., 2024; Qwen et al., 2024; Bai et al., 2025) have achieved tool invocation proficiency that rivals proprietary frontiers like GPT-4 (Achiam et al., 2023) and Gemini (Team et al., 2023). As illustrated in Figure 1, under the prevailing *static world assumption*, where the distribution of tools, schemas, and interaction logic remains consistent between training and inference (Shen et al., 2023; Yin et al., 2025; He et al., 2025), both SFT and RL paradigms demonstrate stable and continuous performance gains, eventually converging on near-perfect success rates.

However, this stability is often an artifact of the closed-set nature of current benchmarks. Real-world deployment is fundamentally non-stationary: APIs are deprecated, tool schemas evolve, and user instructions are frequently ambiguous. When these environmental dynamics diverge from training priors, the apparent mastery observed in static benchmarks often proves fragile. This discrepancy raises a fundamental research question: ***Do current training paradigms enable agents to generalize to the open world?***

To rigorously address this, we formally define **OpenAgent** (Tool-Use Agent in Open-World), a problem setting characterizing shifts across four dimensions: *User Queries* ($\Delta\mathcal{Q}$), *Tool Sets* ($\Delta\mathcal{A}$), *Interaction Dynamics* ($\Delta\mathcal{O}$) and *Task Domain* ($\Delta\mathcal{D}$). To isolate the impact of these shifts from the inherent instability and transient noise of real-world APIs, we establish a controlled sandbox environment. This setup allows for controlled probing, enabling us to maintain a pure closed-set baseline while systematically injecting open-world perturbations across a four-tier diagnostic framework: *Perception*, *Interaction*, *Reasoning*, and *Internalization*.

Our comprehensive evaluations reveal varying degrees of generalization and adaptability in SFT and RL models under open-world settings, while further identifying their respective structural weaknesses. We find that **both** paradigms suffer from performance degradation in open environments, albeit through distinct failure modes. While SFT agents are prone to trajectory overfitting and brittle symbolic anchoring, RL agents, despite demonstrating better semantic

---

[*]Equal contribution  [1]School of Intelligence Science and Technology, Nanjing University, Nanjing, China [2]National Key Laboratory for Novel Software Technology, Nanjing University, Nanjing, China. Correspondence to: Lan-Zhe Guo <guolz@lamda.nju.edu.cn>.

*Proceedings of the 43$^{rd}$ International Conference on Machine Learning*, Seoul, South Korea. PMLR 306, 2026. Copyright 2026 by the author(s).

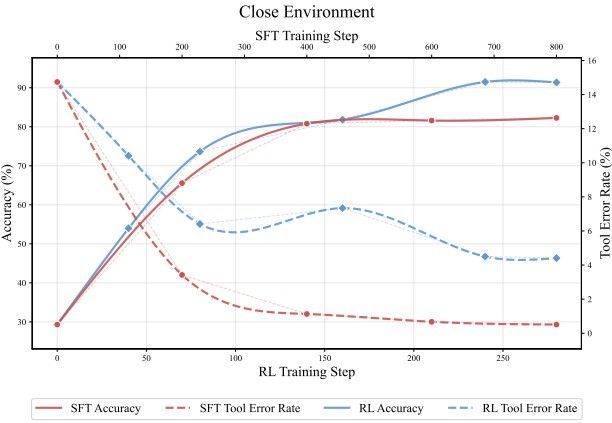

*Figure 1.* Performance evolution of SFT and RL agents in the closed setting. Both approaches show stable and continuous improvement throughout the training process.

grounding, remain vulnerable to boundary blindness due to a teleological bias in their reward structures.

Building on these insights, we introduce Perturbation-Augmented Fine-Tuning (PAFT), a data-centric intervention strategy for SFT. By injecting controlled observation anomalies and symbolic noise into training trajectories, PAFT decouples models from shallow pattern memorization and restores robust generalization in open worlds.

In summary, the primary threefold contributions of this work are as follows:

- **Formalization of Open Environment in Tool Use:** We formally define the OpenAgent setting, characterizing the shifts in queries, tool sets, interaction dynamics and task domain inherent to real-world deployments.

- **Systematic Evaluation in Open Environments:** We establish a controlled sandbox environment and conduct a comprehensive series of experiments to evaluate the robustness of SFT and RL agents across varying levels of environmental shifts.

- **Key Insights and Methods:** We identify critical failure modes in current paradigms, specifically symbolic anchoring in SFT and boundary blindness in RL, and propose PAFT to effectively mitigate performance degradation.

**Conflict of Interest Disclosure.** The authors declare no conflicts of interest.

## 2. Related Works

**Post-training Paradigms for Agents** Current methodologies are categorized into three methods. SFT-based works like GenTool (He et al., 2025), TOUCAN (Xu et al., 2025b), and others (Shi et al., 2024; He et al., 2025; Yuan et al., 2025;

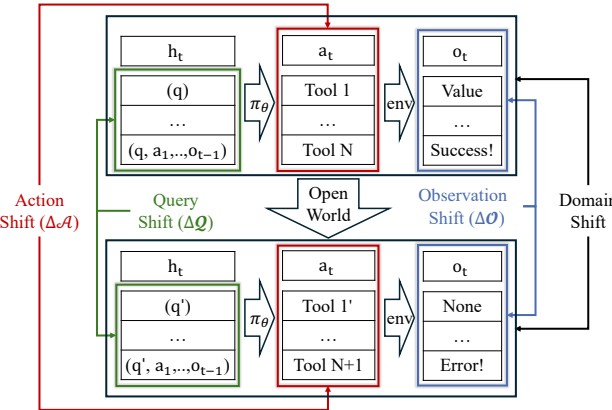

*Figure 2.* Challenges in OpenAgent setting, including query, action, observation and domain shifts.

Lin et al., 2025; Mo et al., 2026) standardize tool invocation via token prediction. Conversely, RL frameworks such as ToolRL (Qian et al., 2025), DeepEyes (Zheng et al., 2026), and others (Feng et al., 2025; Yu et al., 2025; Qian et al., 2025) utilize reward mechanisms to drive robust decision-making. Hybrid paradigms including ReTool (Feng et al., 2026), DeepEyesV2 (Hong et al., 2026), and others (Su et al., 2025; Li et al., 2026) combine SFT initialization with RL exploration. Following recent closed-source advances models like Deepseek-V3.2, GLM-4.5 (Zeng et al., 2025), and Kimi K2 (Team et al., 2025), we select full-parameter SFT and RL as baselines.

**Evaluation Paradigms for Agents** Agent capabilities are evaluated via local tools (Wang et al., 2024a; Guo et al., 2024; Huang et al., 2024; Ye et al., 2025) and remote protocols (Li et al., 2023; Wang et al., 2024c; Gao et al., 2025; Wang et al., 2026). Existing benchmarks mainly prioritize task success (Mialon et al., 2024; Liu et al., 2025; Patil et al., 2025; Shao et al., 2026), while robustness studies focus on inference-time perturbations (Ye et al., 2024; Xiong et al., 2025). Recently, agent benchmarks further move toward realistic environments (Liu et al., 2024; Zhou et al., 2024; Drouin et al., 2024; Jimenez et al., 2024; Koh et al., 2024; Trivedi et al., 2024; Ma et al., 2024; Xie et al., 2024; Xu et al., 2025a). Unlike open-environment learning for passive models (Guo et al., 2020; Li et al., 2021; Zhou, 2022; Guo & Li, 2022; Jia et al., 2024; Guo et al., 2025; Cheng et al., 2025), OpenAgent studies shifts in multi-step. While prior work (Chu et al., 2025) shows RL's advantage over SFT, it lacks tool-use contexts; we address this gap by analyzing training-to-testing transfer under open-environment shifts.

## 3. Preliminaries

### 3.1. Problem Formulation

We model the agent as a sequential decision-making process $\mathcal{M} = \langle \mathcal{Q}, \mathcal{A}, \mathcal{O}, \pi \rangle$, comprising user queries $\mathcal{Q}$, tool space

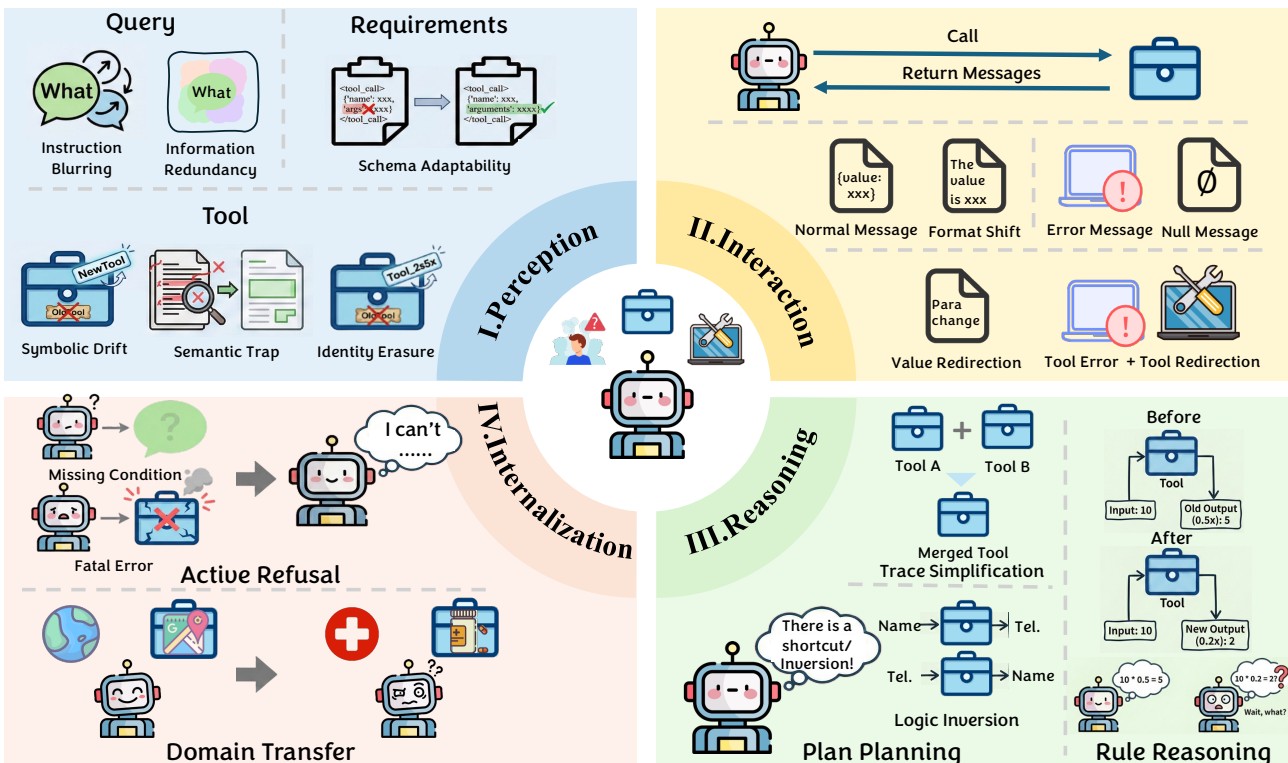

*Figure 3.* Architecture Diagram of the Evaluation Task. We partition this evaluation task architecture diagram into four levels from shallow to deep: Perception, Interaction, Reasoning, and Internalization.

$\mathcal{A}$, and observation space $\mathcal{O}$. Given a query $q \in \mathcal{Q}$, at each step $t$, the agent uses policy $\pi_\theta(a_t \mid h_t)$ to select action $a_t \in \mathcal{A}$ based on history $h_t = (q, a_1, o_1, \ldots, a_{t-1}, o_{t-1})$. The environment returns observation $o_t \in \mathcal{O}$, which updates the history to $h_{t+1} = (h_t, a_t, o_t)$ until task completion. Our core challenge is the generalization of the model when test-time distributions diverge from training priors.

### 3.2. The OpenAgent Setting: Distributional Shifts

Unlike distributional shifts in passive models, where each input is perturbed independently, tool-use agents face *cascading* shifts along the agent-environment loop. A perturbation at step $t$ may alter observation $o_t$, which enters the history $h_{t'}$ and conditions all subsequent decisions $\pi_\theta(a_{t'} \mid h_{t'})$ for $t' > t$, causing errors to compound along the trajectory. As illustrated in Figure 2, OpenAgent covers query shifts $(q \rightarrow q')$, action-space shifts including perturbed existing tools and newly introduced tools, observation shifts such as *None* or *Error*, and joint domain shifts where all interaction elements change together. We decompose this interactive shift space into four dimensions.

**Intent Shift in Query Space** ($\Delta\mathcal{Q}$). The intent distribution $P(q)$ shifts across both semantic directives and linguistic surface forms. Unlike passive input perturbations, query misinterpretation in tool-use agents propagates through the

trajectory: an initial misparse may seed a misaligned tool chain whose errors compound across subsequent steps.

**Structural Shift in Action Space** ($\Delta\mathcal{A}$). The tool space becomes non-stationary ($\mathcal{A}_{\text{test}} \neq \mathcal{A}_{\text{train}}$), spanning multiple levels of change: *surface drift*, which renames identifiers while preserving functionality; *semantic conflict*, which retains familiar names but alters documented behavior; and *structural reconfiguration*, which merges, splits, or inverts tool dependencies. The agent must distinguish cosmetic changes from fundamental reconfigurations and ground actions in the current tool semantics.

**Dynamics Shift in Observation Space** ($\Delta\mathcal{O}$). The feedback channel becomes non-stationary: test-time observations may include return formats, error states, null values, or corrective instructions that are absent or rare in training. Unlike static outputs, these observations directly update the agent's belief state and reshape the subsequent action space. Crucially, the agent must discriminate *informative* anomalies that require policy adjustment from *terminal* failures that require task refusal, a distinction absent in passive settings where each input simply calls for a prediction.

**Compositional Domain Shift** ($\Delta\mathcal{D}$). This shift jointly changes $\mathcal{Q}$, $\mathcal{A}$, and $\mathcal{O}$ into a new domain while preserving a latent problem-solving structure $\mathcal{G}$. Despite distinct

surface semantics and interfaces, the agent must transfer the underlying reasoning topology by re-grounding its nodes and dependencies in the target domain, rather than relying on domain-specific lexical or procedural patterns.

# 4. Evaluation Tasks

## 4.1. Experimental Setup

**Controlled Sandbox Environment.** We utilize a synthetic sandbox environment to enable controlled probing of agent capabilities. Unlike real-world APIs which introduce uncontrollable transient noise, a sandbox allows us to precisely isolate environmental variables and establish a pure closed environment baseline to measure open environment shifts. We synthesized a diverse set of tool-use tasks focused on simple geographical POI queries and calculation problems to test reasoning boundaries, the full specifications of which are detailed in the Appendix A.

**Datasets and Protocols.** The dataset comprises 6,050 training samples and 880 evaluation samples. To ensure rigorous evaluation, we enforced a strict separation between the datasets where no variation patterns from the test set appear during the training phase. This guarantees that the reported performance reflects genuine robustness rather than information leakage or pattern memorization.

**Training Implementation.** We employ Qwen2.5-7B-Instruct (Qwen et al., 2024) as the backbone model. The SFT stage uses full-parameter fine-tuning, while the RL stage adopts the GRPO framework with sparse answers and format rewards. To further ensure a fair comparison with SFT, the RL environment provides only basic error codes upon failure without detailed guidance. We trained both models under comparable computational budgets until they achieved convergence in the closed environment as illustrated in Figure 1. The Training details are provided in Appendix B.

**Metrics.** We employ four metrics to assess performance across correctness, efficiency, robustness, and adaptability:

- **Tool Error Rate (TER):** Measures *robustness* via the ratio of invalid invocations to total calls ($TER = N_{error}/N_{total}$). A lower TER indicates precise adherence.

- **Active Exploration Score (AES):** Quantifies *adaptability* by penalizing "cognitive inertia" when tools fail.

$$AES = e^{-(n_{old}-1)} \tag{1}$$

where $n_{old}$ is the count of repeated invalid attempts. A score of 1.0 signifies immediate adaptation.

- **Average Tool Chain Length (ATCL):** Gauges *efficiency* by averaging tool invocations on solvable tasks. Lower values reflect concise reasoning topologies.

- **Refusal Rate (RR):** Evaluates *boundary awareness* in unsolvable scenarios by measuring the ratio of correctly identified unsolvable cases ($RR = N_{refusal}/N_{unsolvable}$). A higher RR indicates superior active boundary detection.

## 4.2. Tier 1: Perception Generalization

In the context of agentic interaction, perception acts as the cognitive gateway grounding raw observations into actionable representations. To successfully initiate a task, an agent must traverse three representational alignments: deciphering unstructured user intents, aligning with rigid interface protocols, and comprehending functional tool definitions. We systematically evaluate the generalization boundaries of these capabilities through the following three dimensions.

**Instruction Robustness ($\Delta\mathcal{Q}$).** Real-world user instructions are rarely canonical. We evaluate whether the agent can extract invariant intents from non-standard inputs by introducing two distinct types of variations.

- **Instruction Blurring:** We simulate scenarios where user intent is obfuscated by syntactic variance (e.g., colloquialisms, rhetorical questions) or semantic ambiguity (e.g., implicit references, omitted entities). This tests the model's ability to reconstruct exact intents without relying on explicit keyword matching.

- **Information Redundancy:** We test the robustness of the model's attention mechanism against information overload. By embedding the core instruction within irrelevant context or verbose background details, we evaluate the agent's capability to filter out red herrings and lock onto functional constraints.

**Schema Adaptability ($\Delta\mathcal{A}$).** Agents often overfit to specific API formats seen during training. In this experiment, we introduce protocol shifts by modifying the reserved keywords in the tool invocation schema. For instance, we alter the parameter dictionary key from the standard `arguments` to variants like `args` or `parameters`. This verifies whether the model comprehends the abstract concept of parameter passing or merely performs rigid template completion based on memorized structural tokens.

**Semantic Grounding ($\Delta\mathcal{A}$).** Current agents often rely on Symbolic Anchoring with rigid dependence on memorized identifiers rather than true understanding of functional definitions. To decouple these factors, we investigate whether models can adapt to dynamic documentation shifts through a Drift-Trap-Erasure trilogy:

- **Symbolic Drift:** We test tolerance to perturbations via two sub-variants: *Noise Injection* (appending non-semantic suffixes) and *Synonymous Rewriting* (using LLMs to paraphrase). This verifies generalization across semantically equivalent but symbolically distinct labels.
- **Semantic Trap:** We engineer a conflict between sym-

bol and semantics with adversarial attack.by retaining a familiar tool name but modify its description to explicitly forbid usage. If the agent invokes it, it proves that symbolic overrides semantic understanding.

- **Identity Erasure:** We perform the ultimate test of grounding by replacing meaningful tool names with random identifiers (e.g., Tool_2A3D). This removes symbolic shortcuts, forcing the agent to rely exclusively on reading comprehension of the documentation to solve the task.

### 4.3. Tier 2: Interaction Generalization

Interaction generalization evaluates whether an agent can use environmental feedback to refine subsequent actions. In open environments, observations may indicate format changes, execution failures, or new operational constraints rather than merely confirming previous tool calls. We examine three capabilities: *Format Adaptability* for parsing diverse observation structures, *Error Correction* for recovering from explicit or silent failures, and *Dynamic Alignment* for adjusting actions according to feedback-provided rules.

**Format Adaptability** ($\Delta\mathcal{O}$). Agents typically overfit to standard JSON formats for parsing observations. This experiment tests the fundamental capability of the agent to parse non-standard feedback. We introduce Format Shift by altering the structure of tool return values from structured JSON to unstructured formats such as Markdown tables or raw text logs. This evaluates whether the agent extracts key information based on semantic content rather than relying on rigid syntactic parsing templates.

**Error Correction** ($\Delta\mathcal{O}$). Open environments may expose agents to execution failures that interrupt the planned trajectory. We evaluate whether the agent can diagnose such failures and recover by revising its invocation plan or selecting alternative tools. We consider two types of failure:

- **Error Return:** The primary tool explicitly returns an error code or traceback message after invocation. This tests whether the agent can interpret explicit failure signals and switch to a redundant tool to resolve the blockage.
- **Null Return:** The primary tool returns a null value or an empty string without any warning. This tests whether the agent can detect an implicit anomaly where the return is syntactically valid but semantically uninformative, and infer that corrective action is required.

**Dynamic Alignment** ($\Delta\mathcal{O}$). Real-world environments are non-stationary and may provide corrective instructions within feedback to steer the agent away from obsolete assumptions. We evaluate whether the agent can perform feedback-conditioned policy adjustment through two active guidance mechanisms:

- **Value Redirection:** The environment notifies the agent

that the valid parameter logic has shifted (e.g., "ID 1024 is deprecated, use ID 2048"). We verify if the agent can grasp this dynamic constraint and reissue the tool call with corrected values.

- **Tool Redirection:** The environment signals that a specific tool schema is obsolete and explicitly suggests an alternative one. This tests whether the agent can abandon its pre-planned trajectory and strictly adhere to the new operational manual provided in the feedback.

### 4.4. Tier 3: Reasoning Generalization

Beyond perception and interaction, *Reasoning* determines whether an agent can compose individual tool calls into a valid execution plan. Instead of triggering APIs sequentially according to memorized trajectories, a robust agent should infer the current rules governing tool behavior and construct an execution graph that satisfies the task dependencies. We evaluate this ability through two complementary dimensions: *Rule Reasoning*, which tests whether the agent can adapt to changed functional definitions, and *Path Planning*, which tests whether it can reorganize tool dependencies under altered execution structures.

**Rule Reasoning** ($\Delta\mathcal{A}$). Models may memorize training-time rules instead of performing deduction from the current tool description. We introduce a *Calculation Shift* by modifying the functional semantics of selected tools while keeping their interfaces unchanged. For example, a tool originally defined to return a value multiplied by $0.5$ is redefined in the test setting to use a factor of $0.1$. This experiment verifies whether the agent can derive the correct result from the updated rule description, or instead reverts to the memorized rule from the training distribution.

**Path Planning** ($\Delta\mathcal{A}$). This dimension evaluates whether the agent can construct valid execution graphs under changed tool dependencies. We test whether it genuinely reasons over the current causal structure among tools, rather than overfitting to fixed tool chains observed during training, through two complementary sub-experiments:

- **Trace Simplification:** We test whether the agent can improve execution efficiency when a shortcut becomes available. Specifically, we introduce a composite tool that functionally replaces a multi-step tool chain used during training. The agent should recognize the new affordance and select the efficient composite tool, rather than rigidly following the previously memorized multi-step trajectory.
- **Logic Inversion:** We test robustness to causal reversal in the tool dependency graph. At test time, we invert the documented information-acquisition order, for example by making the output of tool $B$ a prerequisite for invoking tool $A$. This verifies whether the agent can resolve preconditions from the current dependency graph, or in-

stead adheres to the fixed topological order learned from training trajectories.

### 4.5. Tier 4: Internalization Generalization

Internalization generalization evaluates whether an agent has learned task-level principles that go beyond local tool invocation patterns. At this level, a robust agent should not only execute valid tool-use trajectories, but also recognize when a task is infeasible under the current constraints and transfer abstract problem-solving structures across domains. We examine this through two capabilities: *Active Refusal*, which tests solvability boundary recognition, and *Domain Transfer*, which tests whether learned reasoning topologies can be reused in semantically different environments.

**Active Refusal ($\Delta\mathcal{Q} \cup \Delta\mathcal{O}$).** A robust agent should determine whether a task is solvable given the available information and executable tools, rather than forcing a plausible but unsupported answer. We evaluate this boundary-awareness ability through two insufficient-condition scenarios:

- **Missing Condition:** We remove necessary conditions from the user query through a human-verified process, making the task theoretically unsolvable due to information deficiency. This tests whether the agent can detect the logical incompleteness of the request and refuse to answer, instead of fabricating missing information to force completion.
- **Fatal Error:** We simulate an unrecoverable failure where a necessary tool returns a fatal error and no alternative tool is available. This tests whether the agent can recognize the resulting operational deadlock and conclude that the task is currently unexecutable, rather than repeatedly retrying or hallucinating a successful result.

**Domain Transfer ($\Delta\mathcal{D}$).** This dimension evaluates whether the agent can transfer internalized problem-solving structure across domains. We introduce a *Holistic Domain Shift*, where user queries, tool definitions, and environmental observations change jointly. For example, we migrate the testing environment from an address-query scenario to a medical-registration scenario. Although the surface semantics and interfaces change substantially, the underlying causal structure is preserved as a latent structural invariance. This experiment tests whether the agent can re-ground the same abstract reasoning topology, such as a search-select-confirm chain, in a novel target domain rather than overfitting to domain-specific lexical or procedural cues.

## 5. Key Observations

Synthesizing the results across the four tiers of generalization, we distill observations regarding the behavioral divergence between SFT and RL in open-world agentic tasks.

> **Tier 1: Tool Perception**
>
> RL achieves semantic grounding through interaction while SFT relies on brittle symbolic anchoring that degrades as fitting increases.

As illustrated in Figure 4, we observe a consistent degradation pattern where robustness declines as training steps increase. Specifically, in the Semantic Trap and Identity Erasure experiments, the SFT model displays a sharp increase in TER accompanied by a significant drop in accuracy. The dashed blue line indicates that SFT frequently invokes the wrong tools when surface-level names are perturbed. Conversely, while RL also experiences some performance decay due to overfitting, it maintains a consistently lower error rate and higher accuracy retention compared to SFT. The RL agent demonstrates superior resistance to adversarial descriptions and effectively ignores misleading name cues.

This disparity stems from SFT's reliance on *symbolic anchoring*, where the model acts as a superficial pattern matcher that collapses once surface tokens are decoupled from their semantics. In contrast, the penalty-driven nature of RL necessitates *semantic grounding*, compelling the agent to scrutinize functional descriptions rather than memorizing rigid name-context associations.

> **Tier 2: Interaction with Guidance**
>
> RL leverages explicit guidance for dynamic policy adaptation, while SFT exhibits trajectory inertia and often hallucinates outcomes under corrective feedback.

As illustrated in Figure 5, we observe a distinct behavioral divergence dependent on feedback clarity. Under ambiguous signals like null returns, both models exhibit performance degradation. Crucially, this failure mode follows a clear trend where the inability to adapt intensifies as training progresses. As the models become more fitted to the training distribution, their capacity to actively explore under uncertainty progressively declines. A significant disparity emerges under explicit tool redirection. Qualitative case studies reveal that SFT tends to interpret observations as successful confirmations, even when the feedback explicitly redirects the agent. Consequently, it ignores the guidance and hallucinates successful observations to force the continuation of its pre-planned trajectory. In contrast, RL maintains high adaptability by parsing the instruction and migrating to the new tool immediately, and then continues reasoning.

We hypothesize that this behavioral divergence stems from differences in training paradigms. SFT is optimized on clean demonstrations where observations consistently validate actions. This likely conditions the model to disregard the causal link between feedback content and the subsequent policy, thereby making its behavior resemble an open-loop

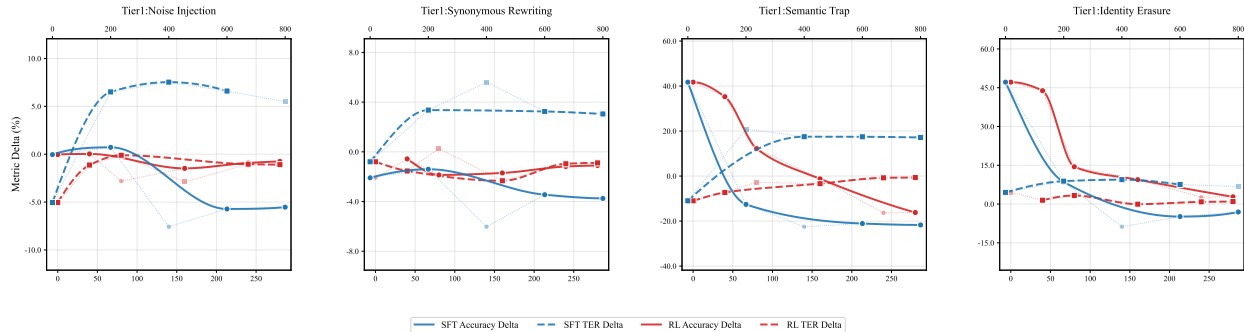

*Figure 4.* Accuracy Delta and TER in Tier-1 Perception. SFT exhibits brittle symbolic anchoring and underperforms compared to RL when tool semantics shift. For detailed case analyses, refer to Appendix E. Delta represents the performance gap relative to the closed-set baseline. Setup and absolute values are provided in Appendix C and D.

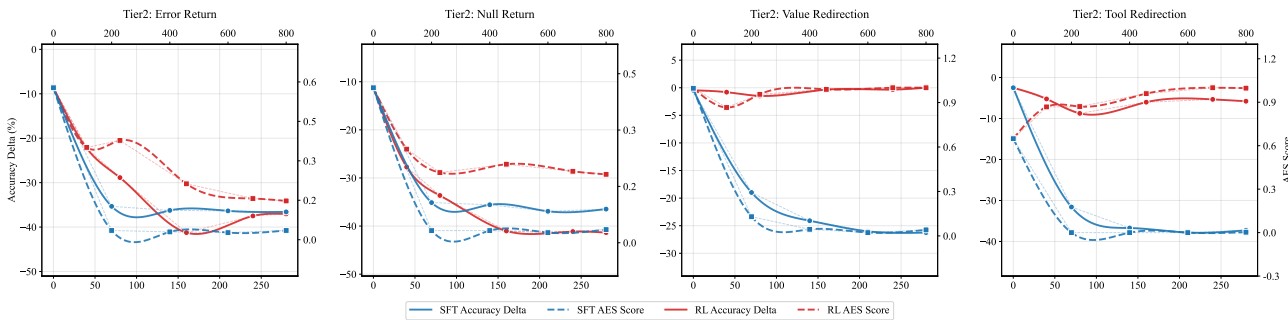

*Figure 5.* Accuracy Delta and AES Score in Tier-2 Interaction. Both RL and SFT degrade under ambiguous feedback, but RL maintains superior resilience under explicit guidance while SFT fails to adapt. Delta represents the performance gap relative to the closed-set baseline. Setup and absolute values are provided in Appendix C and D.

execution policy. Conversely, RL encourages a more closed-loop behavior because ignoring error signals is penalized during interaction. This interaction mechanism potentially forces the agent to treat environmental observations as critical state updates, ensuring that explicit environmental instructions are prioritized to maximize reward.

> **Tier 3: Reasoning Generalization**
>
> RL demonstrates generalization in local adaptation, yet both paradigms suffer substantial degradation under global dependency inversion.

As illustrated in Figure 6, RL exhibits superior generalization in local optimization tasks. In *Trace Simplification*, the RL model achieves the lowest ATCL, indicating it successfully abandons the memorized training-time sequence to utilize efficient composite tools. Similarly, in *Rule Reasoning*, RL outperforms SFT by dynamically adhering to updated calculation logic, whereas SFT shows a stronger tendency to regress to memorized training constants.

However, a critical shared failure emerges in *Logic Inversion*. When the causal dependency between tools is reversed (e.g., shifting from $A \rightarrow B$ to $B \rightarrow A$), both SFT and RL suffer precipitous performance drops. Qualitative anal-

ysis in Appendix E reveals that despite the documentation explicitly stating the new prerequisite order, both models frequently hallucinate the existence of the training-set topology. This suggests that while RL generalizes well to local rule shifts, both paradigms overfit to the temporal order of action sequences, treating the dependency graph as a fixed action order rather than a dynamically specified logical structure.

> **Tier 4: Boundary Awareness**
>
> Both paradigms struggle with boundary awareness in unsolvable states, but exhibit distinct failure patterns: SFT often fails to perceive fatal feedback, whereas RL perceives the failure but still favors forced completion.

As illustrated in Figure 7, both models struggle to identify unsolvable queries, leading to low refusal success rates when tools are unavailable. However, a fine-grained analysis of non-refusal cases under *Fatal Error* scenarios reveals that SFT and RL fail through qualitatively distinct mechanisms. Case studies in Appendix E.4.1 show that upon receiving an error, SFT directly outputs an unsupported answer without acknowledging the error at all, effectively treating the error observation as if it were a normal return value. In contrast, RL explicitly recognizes the failure (e.g., "It seems the

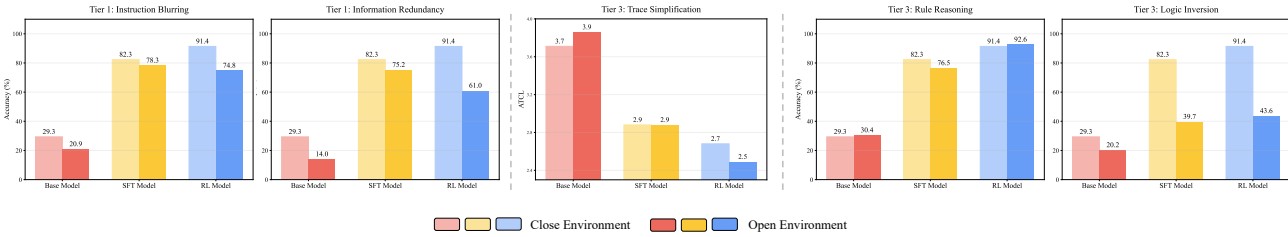

*Figure 6.* Accuracy comparison on Tier 1-Instruction Robustness (left) and Tier 3-Rule Reasoning & Path Planning (right), evaluated at the stable training phase. While SFT models exhibit degradation across all perturbations, RL models show drops primarily under logic inversion and query variations. Detailed setups and full training dynamics are provided in Appendices C and D.

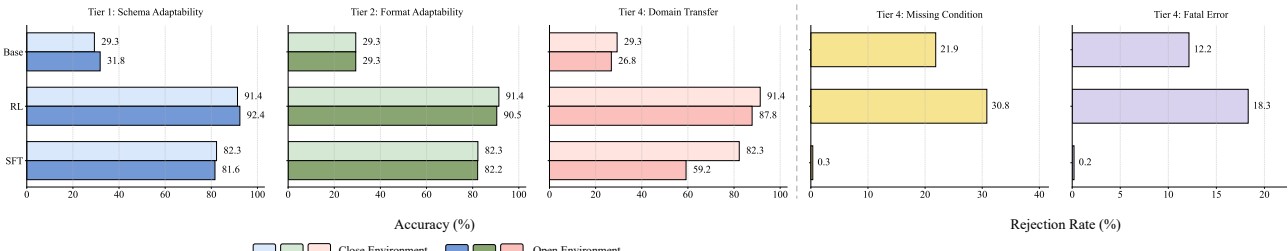

*Figure 7.* Accuracy in Tier 1-Schema Adaptability, Tier 2-Format Adaptability, and Tier 4-Domain Transfer (left) and Refusal Rate in Tier 4-Active Refusal (right), evaluated at the stable training phase. Both SFT and RL models are robust to simple format changes, whereas SFT shows significant degradation under domain transfer. Both methods demonstrate limited active refusal for unsolvable queries. Detailed setups and full training dynamics are provided in Appendices C and D.

tool is not available"), demonstrating a notably high error awareness rate, yet it still proceeds to fabricate an answer rather than issuing a refusal.

This divergence is consistent with the open-loop versus closed-loop dichotomy observed in earlier tiers. SFT's failure is perceptual: its open-loop policy does not parse environmental feedback, so the error signal is simply ignored. RL's failure, however, is more decisional: it perceives the error but still favors completion, which is consistent with a *teleological bias* induced by outcome-oriented rewards. Because training rewards presuppose that every problem is solvable, the policy learns to provide an answer at all costs, prioritizing completion over honest refusal. This is distinct from mere overconfidence; it reflects a structural limitation of outcome-based reward design that lacks an explicit incentive to abstain when the task is infeasible. Such persistent forced-completion behavior raises critical agent safety concerns, as models may sacrifice truthfulness for helpfulness in unpredictable open-world deployments.

## 6. Perturbation-Augmented Fine-Tuning

Our analysis reveals that SFT's generalization decay stems from a distributional mismatch: standard SFT optimizes $\mathcal{L} = -\mathbb{E}_{(h,a)\sim\mathcal{D}_{\text{clean}}}[\log \pi_\theta(a|h)]$, where all training histories $h$ are drawn exclusively from successful executions. This confines the learned policy to a narrow manifold $\mathcal{H}_{\text{clean}}$ that excludes error states, ambiguous feedback, and symbolic variations. To bridge this gap, we propose

**Perturbation-Augmented Fine-Tuning (PAFT)**, a data intervention strategy that augments $\mathcal{D}_{\text{clean}}$ with controlled trajectory-level perturbations:

$$\mathcal{D}_{\text{PAFT}} = (1-\alpha)\,\mathcal{D}_{\text{clean}} \,\cup\, \alpha \cdot \mathcal{G}(\mathcal{D}_{\text{clean}}) \qquad (2)$$

where $\mathcal{G}$ applies perturbation operators that expand the support of $h$ into regions containing anomalous observations, unsolvable states, and symbolic variants. Unlike standard augmentation that perturbs inputs in isolation, PAFT operates at the trajectory level, injecting perturbations mid-trajectory and supervising the corrective continuations.

### 6.1. Method

**Environmental Feedback Perturbation (EFP).** To combat the open-loop trajectory inertia, we extend successful sub-trajectories by injecting stochastic observation anomalies that simulate unexpected environmental feedback. A standard chain is transformed via a generator $\mathcal{G}_{env}$ as:

$$\tau_{\text{orig}} = \{a_i, o_i\} \xrightarrow{\mathcal{G}_{env}} \tau' = \{a_i, o_{\text{change}}, a'_i, o_i\} \qquad (3)$$

Here, $o_{\text{change}}$ represents a feedback anomaly (e.g., tool redirection or value deprecation). The model is supervised to generate a corrective action $a'_i$ that recovers from the disturbance, rather than blindly assuming success. This converts the open-loop pattern into a closed-loop recovery behavior.

**Solvability Boundary Perturbation (SBP).** To address the boundary blindness identified in Tier 4, we construct negative samples that enforce explicit refusal behavior. A refusal

*Table 1.* PAFT results across open environment tiers.

| Model | Tier-1 Acc Δ↑ | Tier-2 Acc Δ↑ | Tier-3 Acc Δ↑ | Tier-4 RR↑ |
|---|---|---|---|---|
| *Performance across training stages* | | | | |
| Base | -29.8 | -8.5 | -8.5 | 12.2 |
| SFT-200 | -67.7 | -48.2 | -39.9 | 0.3 |
| + PAFT | **+28.6** | **+26.5** | **+22.7** | **99.3** |
| SFT-400 | -53.9 | -45.4 | -32.5 | 0.0 |
| + PAFT | **+5.6** | **+4.9** | -2.8 | **97.8** |
| SFT-600 | -51.3 | -46.4 | -33.0 | 0.1 |
| + PAFT | -2.5 | -2.9 | -10.7 | **99.6** |
| SFT-800 | -50.4 | -45.3 | -28.0 | 0.2 |
| + PAFT | -4.1 | -5.3 | -9.8 | **99.6** |
| *Ablation: perturbation ratio $\alpha$ (at 800 steps)* | | | | |
| $\alpha = 0.2$ | -6.9 | -7.7 | -15.8 | 90.4 |
| $\alpha = 0.3$ (default) | **-4.1** | **-5.3** | **-9.8** | **99.6** |
| $\alpha = 0.4$ | -5.3 | -7.8 | -11.1 | 96.4 |
| *Ablation: individual components (at 800 steps, $\alpha = 0.3$)* | | | | |
| EFP only | -30.1 | **-3.2** | -27.0 | 1.7 |
| SBP only | -38.9 | -36.5 | -31.7 | **99.2** |
| SRP only | **-14.9** | -34.2 | **-10.6** | 2.1 |

generator injects unrecoverable errors into the observation, modifying the trajectory to:

$$\tau_{\text{orig}} = \{a_i, o_i, a_{i+1}\} \xrightarrow{\mathcal{G}_{\text{refusal}}} \tau' = \{a_i, o_{\text{fatal}}, a_{\text{end}}\} \quad (4)$$

where $o_{\text{fatal}}$ signifies a fatal error with no alternative tools available, and $a_{\text{end}}$ represents the explicit refusal action. This breaks the implicit "every problem is solvable" assumption in $\mathcal{D}_{\text{clean}}$ and teaches the model to halt execution rather than hallucinate.

**Symbolic Representation Perturbation (SRP).** To decouple the symbolic anchoring identified in Tier 1, we apply surface-level augmentations to the tool definition space. An auxiliary agent performs synonymous rewriting and noise injection on tool names and documentation content. This forces the model to ground its decision-making in invariant functional semantics rather than memorized token sequences. To ensure evaluation fairness, all perturbation patterns are automatically generated without any prior knowledge of the specific variation types used in the test set, reducing the risk of information leakage.

### 6.2. Experimental Results

We evaluate PAFT on the most challenging tasks from each tier: *Semantic Trap* (Tier-1), *Tool Redirection* (Tier-2), *Logic Inversion* (Tier-3), and *Fatal Error* (Tier-4). For Tiers 1–3, we report the Accuracy Delta ($\Delta$) relative to the closed-set baseline; for Tier-4, we report the absolute RR. We set $\alpha = 0.3$ as the default perturbation ratio.

**Performance.** As shown in the upper portion of Table 1,

PAFT yields consistent improvements across all training stages and tiers. For early-stage models, PAFT transforms severe degradation into positive deltas across Tiers 1–3, indicating that perturbation-augmented training can preemptively prevent overfitting to clean trajectories.

**Ablation Study.** The lower portion of Table 1 reveals two key findings.

*(1) Perturbation ratio.* Increasing $\alpha$ from 0.2 to 0.3 yields clear gains across all tiers, while further increasing to 0.4 provides marginal or no additional benefit. We attribute this diminishing-return pattern to a trade-off between distributional coverage and signal quality: at low $\alpha$, the model is exposed to too few perturbed trajectories to learn robust recovery behaviors; as $\alpha$ grows, the augmented data progressively covers more failure regions, but beyond a saturation point, an excessive proportion of perturbed samples dilutes the clean supervision signal and introduces conflicting gradients between normal task completion and anomaly handling, leading to no further improvement or even slight regression. The result suggests that $\alpha{=}0.3$ achieves a near-optimal balance between robustness and task performance.

*(2) Component contributions.* EFP improves Tier-2 interaction adaptability, as the injected observation anomalies teach the model to parse and react to unexpected feedback rather than executing blindly along pre-planned trajectories. Notably, EFP contributes minimally to Tier-4 refusal, indicating that error adaptation and unsolvable-state recognition capture distinct capabilities. SBP almost exclusively restores Tier-4 refusal capability by introducing fatal error samples that break the implicit solvability assumption, while leaving other tiers largely unaffected, which suggests that, in our setting, boundary blindness is strongly tied to data coverage problem where the model never learns to refuse simply because $\mathcal{D}_{\text{clean}}$ contains no refusal demonstrations. SRP primarily enhances Tier-1 perception robustness and Tier-3 reasoning generalization, as symbolic perturbations force the model to attend to functional descriptions rather than surface-level identifiers, mitigating both symbolic anchoring and the associated downstream reasoning errors. This targeted effectiveness, where each component is necessary yet individually insufficient, confirms that PAFT is a structured, diagnosis-driven intervention rather than generic data augmentation.

## 7. Conclusion

In this paper, we formalize the **OpenAgent** setting and leverage a controlled sandbox to diagnose the open-world adaptability of LLM agents. Our systemic analysis reveals that both SFT and RL paradigms exhibit vulnerabilities when facing non-stationary environmental shifts. Furthermore, we propose PAFT, a disturbance-based intervention strategy designed to mitigate trajectory overfitting in SFT.

## Acknowledgements

This work was supported by the Key Program of Jiangsu Science Foundation (BK20243012), the NSFC (62306133), and the 111 Center (No. B26023).

## Impact Statement

This work studies robustness and failure modes of tool-use agents under open-world shifts. Its primary positive impact is to improve the reliability and safety of deployed agents by identifying conditions under which models ignore feedback, misuse tools, or hallucinate answers. Potential risks include misuse of stronger tool-use agents in automated decision-making systems. We mitigate this by emphasizing refusal behavior, boundary awareness, and controlled evaluation rather than unrestricted deployment.

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

# Appendix Contents

## A. Sandbox Environment

Since live APIs and MCP-based tool ecosystems are often unstable, version-dependent, and difficult to control during evaluation, we build a local code-running sandbox to simulate tool calls. Specifically, we create a Python sandbox environment class that includes database, tools implementation, and tool call module. Detailed information is as follows:

### A.1. Database

We built our experimental database using public Point of Interest (POI) data from Beijing. Each POI entry has five key attributes: Phone Number, Address Name, Adname (Administrative Name), Coordinates (Latitude/Longitude), and Category. To strictly prevent data leakage and ensure the model relies entirely on tool usage rather than parameter memory to answer, we adopted an anonymization protocol. Specifically, all Address Names and Phone Numbers are replaced with unique, randomly generated alphanumeric strings and non-repeating number sequences, respectively. In addition, we strictly separate the training and test databases to avoid the model guessing the answers correctly by chance. Examples of the database data are shown in Table 2.

*Table 2.* Samples of the Anonymized POI Knowledge Base

| Address Name | Phone Number | Adname | Coordinates | Category |
|---|---|---|---|---|
| exotlwnk | 00000000 | chaoyang | 116.413486, 39.992582 | C1 |
| cqhecbvi | 00000001 | haidian | 116.318014, 40.029475 | C2 |
| rjztnhhg | 00000002 | changping | 116.303367, 40.160994 | C2 |

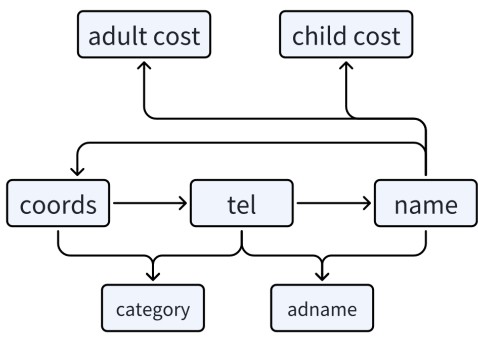

*Figure 8.* Logic Diagram of Tools. Each line represents one tool. The start of an arrow denotes the input, while the end of an arrow denotes the output.

### A.2. Logical Architecture and Function Classification of Tools

We designed the logical architecture of the tools in accordance with the logic diagram (Figure 8) and the following rules:

1. For key-value (coords, tel, name) query functions: we only support the counterclockwise query method by default, i.e., coords can only be used to query tel;

2. For the fare query function: we only allow queries by name;

3. For the distance calculation function: we only allow latitude and longitude as input parameters;

4. For non-key information queries: we allow any three key information to be used as input.

The above rules also restrict the input type of each tool, and we finally obtained 12 tools, which are shown in Table 3.

*Table 3.* Definitions of training set tools with categorized input and output types.

| Tool Name | Functional Description | Input | Output |
|---|---|---|---|
| search_map_coordinates | Retrieve location coordinates by exact name. | Name | Coords |
| administrativedivisionquery | Retrieve the district name (adname) by exact name. | Name | District |
| fetch_poi_category | Retrieve the category type by exact name. | Name | Category |
| find_contact_at_coords | Retrieve the telephone number using coordinates. | Coords | Tel |
| reverse_geocoding | Retrieve the district name using coordinates. | Coords | District |
| venue_type_identifier | Retrieve the category type using coordinates. | Coords | Category |
| caller_id_lookup | Retrieve the name of a POI using its telephone number. | Tel | Name |
| regional_mapping_by_tel | Retrieve the district name using telephone number. | Tel | District |
| service_catalog_search | Retrieve the category type using telephone number. | Tel | Category |
| calculate_distance_by_coords | Calculate distance between two coordinates. | Coords × 2 | Distance |
| get_adult_cost_by_name | Calculate discounted adult price (0.5x) by name. | Name | Price |
| get_child_cost_by_name | Calculate discounted child price (0.2x) by name. | Name | Price |

### A.3. Tool Call Module

We set up the different tool call modules for RL training and evaluation testing in two different ways:

- For RL training, we use the Verl Tool framework. The core idea is to run the code sandbox as a service on the server and open an external port. This port receives tool names and corresponding parameters, then sends back the tool call results. During each RL training run, we use the terminator <tool_call >to detect tool call commands. Specifically, when the model outputs this keyword, it stops generating text and triggers the tool detector. After processing all data in the same batch, the tool detector accesses the port uniformly to get results. These operations ensure only one tool call

command is formed per conversation turn. The rest of the reinforcement learning process is the same as normal RL training.

- For evaluation testing, we simulate multi-turn conversations and also detect tool calls via the terminator, which ensures that only one tool call is triggered in a single turn. If the model outputs $</answer>$, it means the response is finished.

In summary, the experimental environment of our study is composed of the query database, the sandbox code execution environment, the training and test sets, as well as the training and testing logic. The case of interaction is shown below.

---

**Interaction case**

```
{
  "role":  "user",
  "content":  "I have the phone number '00001433'.  Can you help me find
the location of the place associated with this number?"
},
{
  "role":  "assistant",
  "content":  "<tool_call>{"tool":  "caller_id_lookup", "args":  {"tel":
"00001433"}}</tool_call>"
},
{
  "role":  "user",
  "content":  "Observation:  {"name":  "sgtdfxvz" }"
},
{
  "role":  "assistant",
  "content":  "There is no direct tool available to retrieve the location
from a phone number.  Therefore, I will first use the phone number to look
up the name of the place.  <tool_call>{"tool":  "search_map_coordinates",
"args":  {"name":  "sgtdfxvz"}}</tool_call>"
},
{
  "role":  "user",
  "content":  "Observation:  {"location":  "116.445179,39.953269" }"
},
{
  "role":  "assistant",
  "content":  "Now that I have the name of the place, I can
proceed to find its location using the name-based lookup tool.
<answer>116.445179,39.953269</answer>"
}
```

---

### A.4. Training and Test Data

Following data preparation in the main paper and the database and tools mentioned above, we employ a reverse-synthesis approach to generate training instances. By randomly sampling initial query conditions and target answers based on our defined tool set, we derive a symbolic reasoning trajectory. Formally, a symbolic trajectory $\mathcal{T}_{sym}$ is defined as:

$$\mathcal{T}_{sym} = \{c, t_1 \rightarrow t_2 \rightarrow \cdots \rightarrow t_k, a\} \tag{5}$$

where $c$ represents the initial condition (e.g., a specific location coordinate), $t_i$ denotes the $i$-th tool invocation in the chain, and $a$ is the final answer. To align these symbolic representations with human interaction patterns, we utilize an LLM

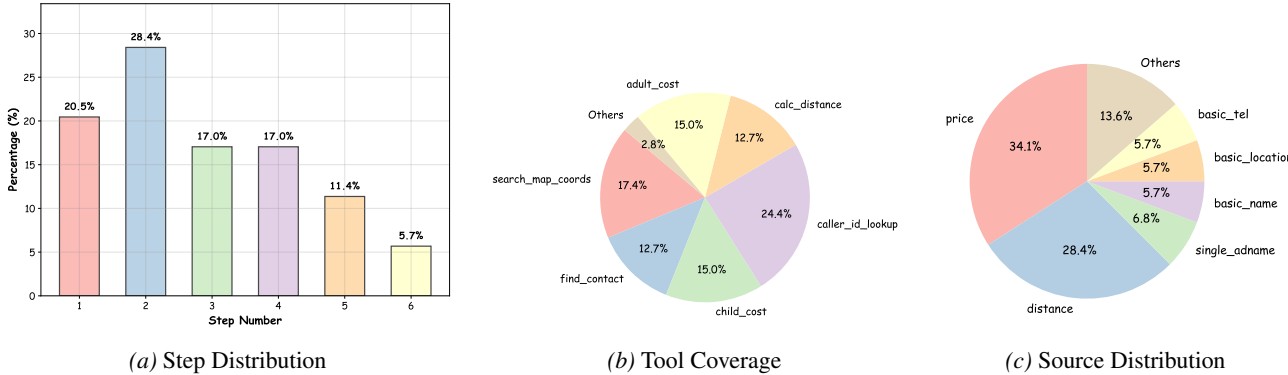

*(a)* Step Distribution          *(b)* Tool Coverage          *(c)* Source Distribution

*Figure 9.* Overall Statistics of the POI Multi-step Tool-calling Dataset: (a) Distribution of task complexity by step count; (b) Proportions of various tool invocations; (c) Composition of task scenarios and sources.

to transform and smooth $\mathcal{T}_{sym}$ into natural language question-answer pairs, resulting in a dataset of coherent multi-step tool-use queries.

The final dataset comprises a total of 6,930 samples, divided into a training set ($N = 6,050$) and a test set ($N = 880$). The dataset characteristics are designed to ensure rigorous evaluation:

- **Complexity Distribution:** Both the training and test sets follow an identical distribution regarding the length of tool chains (e.g., 1-hop, 2-hop, ..., $k$-hop). This ensures that the test set accurately reflects the structural complexity of the training data.

- **Database Isolation:** To eliminate the possibility of data leakage, the training and test sets are synthesized from disjoint subsets of the underlying POI database. Consequently, the model must generalize its reasoning capabilities to unseen entities rather than recalling specific POI attributes seen during training.

Regarding task complexity distribution, the dataset employs a stepped logical chain design covering a range from one to six steps; two-step tasks serve as the core backbone (accounting for 28.4%), while nearly 40% of the samples consist of high-difficulty reasoning tasks involving three or more steps, ensuring depth in the model's learning of long-range tool invocation paths.

In terms of tool and scenario distribution, the dataset exhibits a high degree of domain focus and balance: the core tool caller_id_lookup represents 24.4% of invocations, while ticket price calculation (adult/child cost) and geographic measurement (coords/distance) tools work in close coordination to form the foundation of task execution. This distribution is further corroborated by the source scenarios, where price-related (34.1%) and distance-related (28.4%) contexts dominate, providing the model with abundant material for numerical reasoning and logical alignment.

Overall, this dataset covers basic attribute queries (e.g., POI names, phone numbers, and administrative divisions) and utilizes complex step distributions to force the model into semantic grounding rather than simple symbolic matching during extreme testing, such as anonymization or logical inversion. With its scientific tool sampling proportions and diverse business scenarios, this dataset serves as an ideal benchmark for training agents equipped with efficient path planning and robust generalization capabilities. In addition, we also provide the tools that can be used in Table 3.

## B. Experimental Details

We adopt Qwen2.5-7B-Instruct as the backbone model and perform full-parameter fine-tuning based on the LLaMA-Factory framework. The AdamW optimizer is employed, with the learning rate and batch size set to 3e-5 and 2, respectively. The training process runs for a total of 800 steps, with a model checkpoint saved every 200 steps. For the RL paradigm, training leverages the GRPO tool calling algorithm implemented on the Verl-Tool platform. Notably, we use the normal GRPO to keep comparison fairly. The AdamW optimizer is also used here, with the learning rate set to 1e-6, the number of deployments configured to 8, and the total training steps set to 280. The PAFT algorithm differs from SFT only in the

training dataset. The reward function was designed as follows:

$$Reward = 0.8 \times Acc + 0.2 \times score_{format}$$

Note that we guarantee that both models are fully fitted and strictly control the amount of input data to ensure the fairness of the comparison.

Testing Setup: During the inference phase, all models were uniformly constrained with identical sampling hyperparameters: the sampling temperature was fixed at 0.0, and the Top-p value was set to 1.0. For modifying the tool settings, we conducted three experiments and analyzed the results comprehensively.

The system prompt template for training and testing is as follows:

---

**System Prompt**

```
 "You are a professional geographic information assistant.  You can answer questions about places, phone
numbers, distances, and ticket prices by calling POI database tools.
   ### Available Tools:
   [
   {"name":  "search_map_coordinates", "description":  "Retrieve the location coordinates (longitude,latitude)
of a POI by its exact name.", "parameters": {"type":  "object", "properties":  {"name":  {"type":
"string"}}, "required":  ["name"]}},
   {"name":  "administrative_division_query", "description":  "Retrieve the district name (adname) of a
POI by its exact name.", "parameters":  {"type":  "object", "properties":  {"name":  {"type":  "string"}},
"required":  ["name"]}},
   {"name":  "fetch_poi_category", "description":  "Retrieve the category type of a POI by its exact name.",
"parameters": {"type":  "object", "properties":  {"name":  {"type":  "string"}}, "required":  ["name"]}},
   {"name":  "find_contact_at_coords", "description":  "Retrieve the telephone number of a POI using its
location coordinates.", "parameters": {"type":  "object", "properties":  {"location":  {"type":  "string",
"description":  "'lon,lat' string"}}, "required":  ["location"]}},
   {"name":  "reverse_geocoding", "description":  "Retrieve the district name of a POI using its location
coordinates.", "parameters":  {"type":  "object", "properties":  {"location":  {"type":  "string",
"description":  "'lon,lat' string"}}, "required":  ["location"]}},
   {"name":  "venue_type_identifier", "description":  "Retrieve the category type of a POI using its
location coordinates.", "parameters":  {"type":  "object", "properties":  {"location":  {"type":  "string",
"description":  "'lon,lat' string"}}, "required":  ["location"]}},
   {"name":  "caller_id_lookup", "description":  "Retrieve the name of a POI using its telephone number.",
"parameters": {"type":  "object", "properties":  {"tel":  {"type":  "string"}}, "required":  ["tel"]}},
   {"name":  "regional_mapping_by_tel", "description":  "Retrieve the district name of a POI using its
telephone number.", "parameters":  {"type":  "object", "properties":  {"tel":  {"type":  "string"}},
"required":  ["tel"]}},
   {"name":  "service_catalog_search", "description":  "Retrieve the category type of a POI using its
telephone number.", "parameters":  {"type":  "object", "properties":  {"tel":  {"type":  "string"}},
"required":  ["tel"]}},
   {"name":  "calculate_distance_by_coords", "description":  "Calculate the distance in meters between two
specific coordinates.  Requires mode=2.", "parameters":  {"type":  "object", "properties":  {"start_lat":
{"type":  "number"}, "start_lon":  {"type":  "number"}, "end_lat":  {"type":  "number"}, "end_lon":  {"type":
"number"}, "mode":  {"type":  "integer", "description":  "Verification code."}}, "required":  ["start_lat",
"start_lon", "end_lat", "end_lon", "mode"]}},
   {"name":  "get_adult_cost_by_name", "description":  "Calculate the discounted adult ticket price (0.5x)
by POI name.  Requires mode=1.", "parameters":  {"type":  "object", "properties":  {"name":  {"type":
"string"}, "mode":  {"type":  "integer", "description":  "Verification code."}}, "required":  ["name",
"mode"]}},
   {"name":  "get_child_cost_by_name", "description":  "Calculate the discounted child ticket price (0.2x)
by POI name.  Requires mode=1.", "parameters":  {"type":  "object", "properties":  {"name":  {"type":
"string"}, "mode":  {"type":  "integer", "description":  "Verification code."}}, "required":  ["name",
"mode"]}}
   ]
   ### Protocol:
   1.  ONE TOOL PER TURN: You MUST output only ONE tool call at a time.  Wait for the observation before
doing anything else.
   2.  Tool Format:  To use a tool, strictly output a JSON object with "name" and "arguments" keys.  Example:
{"name":  "get_tel_by_name", "arguments":  {"name":  "Beijing_Hotel"}}
   3.  Final Answer:  Wrap the result in <answer></answer>.  Keep numerical answers to 1 decimal place."
```

---

# C. Implementation Details of OpenAgent Evaluation Task

## C.1. Tier 1: Perception Generalization

### C.1.1. INSTRUCTION ROBUSTNESS.

- **Instruction Blurring:** We employ API calls to rewrite user queries, introducing complexities through syntactic variations (e.g., colloquialisms, rhetorical questions) or semantic ambiguities (e.g., implicit references, entity omissions).
- **Information Redundancy:** We employ API calls to rewrite user queries by embedding core instructions within irrelevant contexts or lengthy background details, thereby simulating real-world informational noise. Please refer to the Appendix E.1.1 for further details.

### C.1.2. SCHEMA ADAPTABILITY.

We alter the parameter dictionary key from the standard `args` to `arguments`, as shown in Appendix E.1.2.

### C.1.3. SEMANTIC GROUNDING.

**Synonymous Rewriting** We assess the model's capability to decouple functional intent from specific symbolic labels by replacing tool identifiers with semantically equivalent alternatives. This requires the model to move beyond simple string matching and demonstrate a deep understanding of the underlying natural language semantics.

*Table 4.* Tool Mapping for **Synonym Rewriting**

| Original Tool Name | → | Synonymous Variant |
|---|---|---|
| `calculate_distance_by_coords` | → | `calculate_separation_via_positions` |
| `search_map_coordinates` | → | `lookup_location_latlong` |
| `get_adult_cost_by_name` | → | `fetch_grownup_price_using_name` |

**Noise Injection** While maintaining the integrity of the tool schema, we introduce subtle syntactic perturbations into the tool names by inserting articles or determiners. We utilize this setup to evaluate the model's tolerance to minor fluctuations in API naming and its stability when encountering non-standardized inputs.

*Table 5.* Tool Mapping for **Surface Noise**

| Original Tool Name | → | Noisy Variant |
|---|---|---|
| `calculate_distance_by_coords` | → | `calculate_the_distance_by_coords` |
| `search_map_coordinates` | → | `search_for_map_coordinates` |
| `get_adult_cost_by_name` | → | `get_1_adult_cost_by_name` |

**Semantic Adversarial** We designed a "description trap" where the familiar tool names are retained, but their functional descriptions are modified to explicitly forbid their previous usage. We require the model to overcome "path dependency" on familiar names, recognize the obsolescence of the original tool based on the updated description, and accurately invoke a new tool with matching functionality.

*Table 6.* Changes in **Semantic Adversarial**

| Status | Tool Name | Functional Role |
|---|---|---|
| Obsolete | `get_adult_cost_by_name` | Service fee calculation only (Forbidden) |
| Target | `fetch_grownup_price_using_name` | Actual ticket price calculation (Correct) |

**Identity Erasure** To isolate the model's reliance on pure functional logic, we replace all human-readable identifiers with randomly generated UUIDs. In this extreme setting, we force the agent to perform reasoning based exclusively on the provided natural language descriptions.

*Table 7.* Tool Mapping for **Identity Erasure**

| Semantic Tool Name | $\rightarrow$ | Anonymous Identifier |
|---|---|---|
| `calculate_distance_by_coords` | $\rightarrow$ | `Tool_a2fb` |
| `search_map_coordinates` | $\rightarrow$ | `Tool_c3d8` |
| `get_adult_cost_by_name` | $\rightarrow$ | `Tool_r6t7` |

## C.2. Tier 2: Interaction Generalization

### C.2.1. FORMAT ADAPTABILITY.

We diversify the structure of tool return values, transitioning from strictly structured JSON to less constrained formats such as Markdown tables or raw text logs.

*Table 8.* Example of **Format Shift** in Tool Observations

| Type | Content Structure |
|---|---|
| **Structured (Old)** | `"content": "Observation: {location: 116.35,40.22} "` |
| **Unstructured (New)** | `"content": "Observation: {location is 116.35,40.22} "` |

### C.2.2. ERROR CORRECTION.

- **Error Return:** We simulate an explicit failure where the primary tool returns a specific error message (e.g., *"This tool is unavailable now"*) or a traceback. This scenario evaluates whether the agent can interpret explicit error signals and proactively switch to a redundant tool to resolve the blockage.

- **Null Return:** We introduce an implicit failure where the primary tool returns a null value or an empty set ($\emptyset$) without any diagnostic warning. This tests the agent's capability to detect anomalies based on the absence of content, rather than the presence of an error signal, and infer the necessity of a fallback execution.

*Table 9.* Mapping of **Tool Failure Modes** and Expected Agent Responses

| Failure Type | Tool Observation (Return) |
|---|---|
| **Error Return** | `{"error": "This tool is unavailable now"}` |
| **Null Return** | `""` |

### C.2.3. DYNAMIC ALIGNMENT.

- **Value Redirection:** We simulate a scenario where the provided parameter logic is deprecated, and the environment returns a message including new, valid parameters.
- **Tool Redirection:** We simulate a scenario where the primary tool returns an error explicitly stating it is obsolete while simultaneously suggesting an alternative tool.

Importantly, we feed the interaction history into the API model (DouBao) to enable it to judge whether the target model refuses to answer.

## C.3. Tier 3: Reasoning Generalization

### C.3.1. RULE REASONING.

We introduce rule conflicts in a special tool by modifying the mathematical constants in the tool description. Specifically, we kept the tool interface unchanged, but significantly adjusted the calculation ratio in the Prompt in the test set to examine whether the model relies on the memory from the training stage or can perform real-time reasoning based on immediate instructions.

*Table 10.* Experimental Setup for **Calculation Shift**: Transitioning from Training Logic to Dynamic Test Logic

| Tool Name | Training rule (Origin) | Test rule (Changed) |
|---|---|---|
| `get_adult_cost` | Discounted Price = **0.5x** Original Price | Discounted Price = **0.1x** Original Price |
| `get_child_cost` | Discounted Price = **0.2x** Original Price | Discounted Price = **0.1x** Original Price |

### C.3.2. PATH PLANNING.

**Trace Simplification** We implement Tool shortcut to evaluate the agent's ability to optimize its reasoning trajectory when more efficient paths become available. In this setup, we condense the functionality of two separate, sequential tools into a single, streamlined tool. This transformation reduces a multi-step dependency chain into a one-step execution.

*Table 11.* Implementation of **Tool Consolidation**

| Original Multi-step Tools | → | Consolidated Tool (Single Step) |
|---|---|---|
| `search_map_coords` + `find_contact_coords` | → | `get_tel_by_name` |
| `find_contact_coords` + `caller_id_lookup` | → | `get_name_by_location` |
| `caller_id_lookup` + `search_map_coords` | → | `get_location_by_tel` |

**Logic Inversion** We simulate causal inversions in the dependency figure 8. During the testing phase, we intentionally reverse the information acquisition order described in the documentation, effectively inverting the preconditions of the tools. For example, if a task originally required Tool A to precede Tool B, we redefine the logic such that Tool A now depends on the output of Tool B.

*Table 12.* Implementation of **Causal Reversal**

| Original Tool Logic | → | Reversed Causal Logic (Test) |
|---|---|---|
| `search_map_coords`: Name → **Coords** | → | `lookup_name_by_coords`: Coords → **Name** |
| `find_contact_coords`: Coords → **Tel** | → | `get_coords_by_tel`: Tel → **Coords** |
| `caller_id_lookup`: Tel → **Name** | → | `search_tel_by_name`: Name → **Tel** |

## C.4. Tier 4: Internalization Generalization

### C.4.1. ACTIVE REFUSAL

We implement Active Refusal by constructing "Missing Condition" and "Fatal Error" scenarios, transitioning tasks from being "inevitably solvable" to "deterministically unsolvable." We utilize this setup to assess the model's capability to recognize task boundaries—specifically, whether it can choose to proactively refuse the request rather than generating hallucinations when faced with information deficiency or execution deadlocks. The error tools include `calculate_distance_by_coords`, `search_map_coordinates`, and `get_adult_cost_by_name`.

*Table 13.* Implementation of **Active Refusal** via Insufficient Conditions

| Scenarios | Original State (Solvable) | Perturbed State (Unsolvable) |
|---|---|---|
| **Missing Condition** | Complete information in User Query | **Randomly excised** key entities/constraints |
| **Fatal Error** | Successful Tool Execution | **Irrecoverable error** with no alternative tools |

### C.4.2. DOMAIN TRANSFER

We implement a Domain Shift by changing the distributions of all three interactive elements: the user query, tool definitions, and environmental observations. Specifically, we transfer the evaluation environment from a geographic address lookup task to a medical registration task. While the surface presentation of the tasks is entirely altered, we retain the latent causal logical structure (Figure 8) invariance.

*Table 14.* One-to-One Tool Mapping for Joint Domain Shift

| Geographic Domain (Source $P_{\text{train}}$) | Medical Domain (Target $P_{\text{test}}$) | Latent Logic |
|---|---|---|
| search_map_coordinates | get_medical_facility_coords | $Name \rightarrow Coords$ |
| administrative_division_query | query_medical_administration | $Name \rightarrow Admin$ |
| fetch_poi_category | fetch_medical_service_type | $Name \rightarrow Category$ |
| find_contact_at_coords | find_emergency_hotline_at_coords | $Coords \rightarrow Tel$ |
| reverse_geocoding | reverse_medical_geocoding | $Coords \rightarrow Admin$ |
| venue_type_identifier | identify_clinic_category | $Coords \rightarrow Category$ |
| caller_id_lookup | lookup_facility_by_hotline | $Tel \rightarrow Name$ |
| regional_mapping_by_tel | query_medical_region_by_tel | $Tel \rightarrow Admin$ |
| service_catalog_search | classify_service_by_tel | $Tel \rightarrow Category$ |
| calculate_distance_by_coords | calculate_ambulance_route_distance | $Math(mode = 2)$ |
| get_adult_cost_by_name | get_checkup_cost_by_name | $CostA(mode = 1)$ |
| get_child_cost_by_name | get_vaccine_cost_by_name | $CostB(mode = 1)$ |

## D. Detailed Experimental Results

### D.1. Tier 1 Experiments

We report the full numerical results for all Tier-1 perception experiments. Tables 15–16 present accuracy and ATCL under instruction-level perturbations (Instruction Blurring and Information Redundancy). Table 17 reports performance under schema keyword changes. Tables 18–22 provide per-tool breakdowns across three semantic grounding experiments (Parameter Drift, Synonymous Rewriting, Noise Injection, Semantic Adversarial, and Identity Erasure), where each tool is evaluated independently to isolate tool-specific failure patterns. Across all settings, SFT exhibits increasing symbolic anchoring as training progresses, while RL maintains more stable accuracy retention.

*Table 15.* Detailed Results of Instruction Blurring

| Model/Step | Acc (Abs) | ATCL Ratio | ATCL (Abs) | ATCL Ratio | ATCL Diff |
|---|---|---|---|---|---|
| base | 20.91 | 71.32 | 3.76 | 101.35 | +0.05 |
| sft-200 | 78.98 | 120.45 | 2.78 | 128.70 | +0.62 |
| sft-400 | 67.39 | 83.40 | 2.70 | 96.09 | -0.11 |
| sft-600 | 78.07 | 95.69 | 2.84 | 99.30 | -0.02 |
| sft-800 | 78.30 | 95.17 | 2.83 | 98.26 | -0.05 |
| rl-40 | 34.43 | 63.78 | 2.14 | 93.86 | -0.14 |
| rl-80 | 58.64 | 79.63 | 2.45 | 95.33 | -0.12 |
| rl-160 | 67.27 | 82.22 | 2.56 | 95.52 | -0.12 |
| rl-240 | 74.89 | 81.86 | 2.55 | 95.15 | -0.13 |
| rl-280 | 74.77 | 81.84 | 2.57 | 95.90 | -0.11 |

*Table 16.* Detailed Results of Information Redundancy

| Model/Step | Acc (Abs) | Acc Ratio | ATCL (Abs) | ATCL Ratio | ATCL Diff |
|---|---|---|---|---|---|
| base | 13.98 | 47.68 | 4.01 | 108.09 | +0.30 |
| sft-200 | 84.66 | 129.11 | 2.98 | 137.96 | +0.82 |
| sft-400 | 67.73 | 83.82 | 3.10 | 110.32 | +0.29 |
| sft-600 | 77.39 | 94.85 | 3.16 | 110.49 | +0.30 |
| sft-800 | 75.23 | 91.44 | 3.22 | 111.81 | +0.34 |
| rl-40 | 38.86 | 71.99 | 2.29 | 100.44 | +0.01 |
| rl-80 | 54.77 | 74.38 | 2.36 | 91.83 | -0.21 |
| rl-160 | 55.68 | 68.05 | 2.57 | 95.90 | -0.11 |
| rl-240 | 61.48 | 67.21 | 2.58 | 96.27 | -0.10 |
| rl-280 | 61.02 | 66.79 | 2.56 | 95.52 | -0.12 |

*Table 17.* Detailed Results of Schema Adaptability

| Model/Step | Acc (Abs) | Acc Ratio | ATCL (Abs) | ATCL Ratio | ATCL Diff |
|---|---|---|---|---|---|
| base | 31.82 | 108.53 | 3.72 | 100.27 | +0.01 |
| sft-200 | 76.59 | 116.81 | 2.55 | 118.06 | +0.39 |
| sft-400 | 82.95 | 102.66 | 2.86 | 101.78 | +0.05 |
| sft-600 | 81.93 | 100.42 | 2.87 | 100.35 | +0.01 |
| sft-800 | 81.59 | 99.17 | 2.88 | 100.00 | +0.00 |
| rl-40 | 61.82 | 114.52 | 2.49 | 109.21 | +0.21 |
| rl-80 | 82.39 | 111.88 | 2.74 | 106.61 | +0.17 |
| rl-160 | 81.48 | 99.58 | 2.70 | 100.75 | +0.02 |
| rl-240 | 92.27 | 100.86 | 2.66 | 99.25 | -0.02 |
| rl-280 | 92.39 | 101.13 | 2.66 | 99.25 | -0.02 |

*Table 18.* Detailed Results of Parameter Drift across three tools. T1: calculate_distance_by_coords; T2: search_map_coordinates; T3: get_adult_cost_by_name.

| Model | Tool | Acc (Abs) | Acc Ratio | Tool (Abs) | Tool Ratio | Tool Diff | Fail Rate | Ratio (Base) |
|---|---|---|---|---|---|---|---|---|
| base | T1 | 25.80 | 87.99 | 3.68 | 99.19 | -0.03 | 20.04 | 152.88 |
| | T2 | 26.14 | 89.15 | 3.74 | 100.81 | +0.03 | 18.11 | 137.89 |
| | T3 | 30.11 | 102.69 | 3.71 | 100.00 | +0.00 | 11.25 | 112.27 |
| sft-200 | T1 | 66.82 | 101.91 | 2.18 | 100.93 | +0.02 | 0.15 | 96.85 |
| | T2 | 66.82 | 101.91 | 2.18 | 100.93 | +0.02 | 0.15 | 97.73 |
| | T3 | 66.93 | 102.07 | 2.19 | 101.39 | +0.03 | 6.80 | 85.91 |
| sft-400 | T1 | 80.80 | 100.00 | 2.80 | 99.64 | -0.01 | 1.34 | 94.51 |
| | T2 | 80.23 | 99.29 | 2.80 | 99.64 | -0.01 | 1.17 | 82.49 |
| | T3 | 80.23 | 99.29 | 2.79 | 99.29 | -0.02 | 0.95 | 100.24 |
| sft-600 | T1 | 81.36 | 99.72 | 2.86 | 100.00 | +0.00 | 0.66 | 89.25 |
| | T2 | 81.36 | 99.72 | 2.86 | 100.00 | +0.00 | 0.41 | 100.00 |
| | T3 | 81.93 | 100.42 | 2.86 | 100.00 | +0.00 | 0.47 | 100.00 |
| sft-800 | T1 | 81.82 | 99.45 | 2.88 | 100.00 | +0.00 | 0.41 | 83.47 |
| | T2 | 81.14 | 98.63 | 2.88 | 100.00 | +0.00 | 0.33 | 100.00 |
| | T3 | 81.82 | 99.45 | 2.88 | 100.00 | +0.00 | 0.46 | 133.33 |
| rl-40 | T1 | 55.00 | 101.89 | 2.31 | 101.32 | +0.03 | 21.60 | 112.32 |
| | T2 | 53.98 | 100.00 | 2.27 | 99.56 | -0.01 | 19.88 | 103.32 |
| | T3 | 54.32 | 100.63 | 2.29 | 100.44 | +0.01 | 6.68 | 94.84 |
| rl-80 | T1 | 73.41 | 99.69 | 2.54 | 98.83 | -0.03 | 19.48 | 151.50 |
| | T2 | 77.50 | 105.24 | 2.58 | 100.39 | +0.01 | 16.46 | 127.98 |
| | T3 | 73.98 | 100.46 | 2.57 | 100.00 | +0.00 | 2.20 | 112.15 |
| rl-160 | T1 | 81.93 | 100.13 | 2.64 | 98.51 | -0.04 | 17.93 | 111.43 |
| | T2 | 81.70 | 99.85 | 2.65 | 98.88 | -0.03 | 15.63 | 97.41 |
| | T3 | 80.91 | 98.89 | 2.66 | 99.25 | -0.02 | 0.11 | 50.11 |
| rl-240 | T1 | 91.59 | 100.12 | 2.69 | 100.37 | +0.01 | 9.96 | 99.13 |
| | T2 | 92.05 | 100.62 | 2.68 | 100.00 | +0.00 | 9.90 | 99.03 |
| | T3 | 91.25 | 99.75 | 2.68 | 100.00 | +0.00 | 0.00 | 100.00 |
| rl-280 | T1 | 91.59 | 100.25 | 2.69 | 100.37 | +0.01 | 9.96 | 100.89 |
| | T2 | 92.05 | 100.76 | 2.68 | 100.00 | +0.00 | 9.73 | 99.01 |
| | T3 | 91.36 | 100.00 | 2.68 | 100.00 | +0.00 | 0.00 | 100.00 |

*Table 19.* Detailed Results of symbolic Drift (Synonymous Rewriting) across three tools (**T1**: `calculate_distance_by_coords` → `calculate_separation_via_positions`; **T2**: `search_map_coordinates` → `lookup_location_latlong`; **T3**: `get_adult_cost_by_name` → `fetch_grownup_price_using_name`).

| Model/Step | Tool | Acc (Abs) | Acc Ratio | ATCL (Abs) | ATCL Ratio | ATCL Diff | TER | TER Ratio |
|---|---|---|---|---|---|---|---|---|
| base | T1 | 27.61 | 94.17 | 3.76 | 101.35 | +0.05 | 11.10 | 110.76 |
|  | T2 | 24.09 | 82.16 | 3.88 | 104.58 | +0.17 | 18.22 | 223.78 |
|  | T3 | 30.00 | 102.32 | 3.72 | 100.27 | +0.01 | 12.49 | 95.28 |
| sft-200 | T1 | 68.52 | 104.50 | 2.19 | 101.39 | +0.03 | 3.10 | 39.21 |
|  | T2 | 57.39 | 87.52 | 2.23 | 103.24 | +0.07 | 16.65 | 263.32 |
|  | T3 | 66.59 | 101.56 | 2.20 | 101.85 | +0.04 | 0.60 | 389.74 |
| sft-400 | T1 | 80.45 | 99.57 | 2.79 | 99.29 | -0.02 | 0.71 | 74.47 |
|  | T2 | 72.61 | 89.86 | 2.82 | 100.36 | +0.01 | 8.67 | 1007.77 |
|  | T3 | 71.25 | 88.18 | 2.92 | 103.91 | +0.11 | 10.85 | 765.61 |
| sft-600 | T1 | 80.45 | 98.60 | 2.86 | 100.00 | 0.00 | 0.47 | 100.00 |
|  | T2 | 76.48 | 93.74 | 2.88 | 100.70 | +0.02 | 6.02 | 799.70 |
|  | T3 | 77.50 | 94.99 | 2.86 | 100.00 | 0.00 | 5.27 | 714.63 |
| sft-800 | T1 | 81.59 | 99.17 | 2.88 | 100.00 | 0.00 | 0.70 | 200.70 |
|  | T2 | 75.45 | 91.71 | 2.86 | 99.31 | -0.02 | 5.16 | 787.93 |
|  | T3 | 78.52 | 95.44 | 2.88 | 100.00 | 0.00 | 4.81 | 980.13 |
| rl-40 | T1 | 53.07 | 98.31 | 2.26 | 99.12 | -0.02 | 7.17 | 101.77 |
|  | T2 | 53.52 | 99.15 | 2.33 | 102.19 | +0.05 | 5.06 | 91.35 |
|  | T3 | 53.64 | 99.37 | 2.41 | 105.70 | +0.13 | 14.38 | 74.76 |
| rl-80 | T1 | 70.91 | 96.29 | 2.56 | 99.61 | -0.01 | 6.89 | 350.51 |
|  | T2 | 75.45 | 102.46 | 2.55 | 99.22 | -0.02 | 0.19 | 11.69 |
|  | T3 | 68.98 | 93.67 | 2.55 | 99.22 | -0.02 | 12.93 | 100.56 |
| rl-160 | T1 | 79.89 | 97.64 | 2.65 | 98.88 | -0.03 | 0.34 | 150.34 |
|  | T2 | 79.89 | 97.64 | 2.63 | 98.13 | -0.05 | 0.19 | 100.84 |
|  | T3 | 80.57 | 98.47 | 2.71 | 101.12 | +0.03 | 14.49 | 90.02 |
| rl-240 | T1 | 91.48 | 100.00 | 2.68 | 100.00 | 0.00 | 0.00 | 100.00 |
|  | T2 | 91.59 | 100.12 | 2.67 | 99.63 | -0.01 | 0.00 | 100.00 |
|  | T3 | 87.84 | 96.02 | 2.65 | 98.88 | -0.03 | 10.63 | 105.74 |
| rl-280 | T1 | 91.36 | 100.00 | 2.68 | 100.00 | 0.00 | 0.00 | 100.00 |
|  | T2 | 91.70 | 100.37 | 2.66 | 99.25 | -0.02 | 0.00 | 100.00 |
|  | T3 | 87.73 | 96.03 | 2.65 | 98.88 | -0.03 | 10.55 | 106.83 |

*Table 20.* Detailed Results of symbolic Drift (Noise Injection) across three tools (**T1**: `calculate_distance_by_coords` → `calculate_the_distance_by_coords`; **T2**: `search_map_coordinates` → `search_for_map_coordinates`; **T3**: `get_adult_cost_by_name` → `get_1_adult_cost_by_name`)

| Model/Step | Tool | Acc (Abs) | Acc Ratio | ATCL (Abs) | ATCL Ratio | ATCL Diff | TER Rate-Sub | TER Ratio |
|---|---|---|---|---|---|---|---|---|
| base | T1 | 28.30 | 96.52 | 3.80 | 102.43 | +0.09 | 9.77 | 97.46 |
| | T2 | 30.34 | 103.48 | 3.77 | 101.62 | +0.06 | 8.01 | 98.34 |
| | T3 | 29.20 | 99.59 | 3.81 | 102.70 | +0.10 | 11.31 | 86.26 |
| sft-200 | T1 | 68.52 | 104.50 | 2.20 | 101.85 | +0.04 | 4.26 | 53.79 |
| | T2 | 66.36 | 101.20 | 2.17 | 100.46 | +0.01 | 5.68 | 89.77 |
| | T3 | 63.98 | 97.58 | 2.29 | 106.02 | +0.13 | 19.86 | 12830.28 |
| sft-400 | T1 | 80.57 | 99.72 | 2.79 | 99.29 | -0.02 | 1.07 | 112.90 |
| | T2 | 80.45 | 99.57 | 2.79 | 99.29 | -0.02 | 0.86 | 99.81 |
| | T3 | 58.64 | 72.57 | 2.82 | 100.36 | +0.01 | 24.05 | 1697.62 |
| sft-600 | T1 | 81.02 | 99.30 | 2.86 | 100.00 | +0.00 | 0.23 | 50.12 |
| | T2 | 82.16 | 100.70 | 2.86 | 100.00 | +0.00 | 0.56 | 74.72 |
| | T3 | 64.43 | 78.97 | 2.87 | 100.35 | +0.01 | 21.01 | 2848.55 |
| sft-800 | T1 | 81.70 | 99.31 | 2.88 | 100.00 | +0.00 | 0.35 | 100.35 |
| | T2 | 81.59 | 99.17 | 2.87 | 99.65 | -0.01 | 0.38 | 57.36 |
| | T3 | 66.93 | 81.35 | 2.88 | 100.00 | +0.00 | 17.28 | 3518.94 |
| rl-40 | T1 | 54.09 | 100.20 | 2.28 | 100.00 | +0.00 | 6.73 | 95.48 |
| | T2 | 54.20 | 100.41 | 2.27 | 99.56 | -0.01 | 5.39 | 97.35 |
| | T3 | 53.75 | 99.57 | 2.35 | 103.07 | +0.07 | 15.73 | 81.76 |
| rl-80 | T1 | 72.50 | 98.45 | 2.52 | 98.05 | -0.05 | 1.07 | 54.39 |
| | T2 | 69.20 | 93.97 | 2.48 | 96.50 | -0.09 | 2.49 | 155.84 |
| | T3 | 70.80 | 96.14 | 2.50 | 97.28 | -0.07 | 15.34 | 119.31 |
| rl-160 | T1 | 80.68 | 98.61 | 2.66 | 99.25 | -0.02 | 0.35 | 151.04 |
| | T2 | 78.07 | 95.42 | 2.64 | 98.51 | -0.04 | 0.37 | 201.50 |
| | T3 | 82.27 | 100.55 | 2.68 | 100.00 | +0.00 | 12.62 | 78.42 |
| rl-240 | T1 | 91.25 | 99.75 | 2.68 | 100.00 | +0.00 | 0.00 | 100.00 |
| | T2 | 89.55 | 97.89 | 2.65 | 98.88 | -0.03 | 0.00 | 100.00 |
| | T3 | 90.91 | 99.38 | 2.69 | 100.37 | +0.01 | 10.37 | 103.18 |
| rl-280 | T1 | 91.14 | 99.76 | 2.68 | 100.00 | +0.00 | 0.00 | 100.00 |
| | T2 | 89.77 | 98.26 | 2.65 | 98.88 | -0.03 | 0.00 | 100.00 |
| | T3 | 90.91 | 99.51 | 2.70 | 100.75 | +0.02 | 9.97 | 100.99 |

*Table 21.* Detailed Results of Semantic Adversarial Experiments Across Three Tools (**T1**: `calculate_distance_by_coords` → `calculate_separation_via_positions`; **T2**: `search_map_coordinates` → `lookup_location_latlong`; **T3**: `get_adult_cost_by_name` → `fetch_grownup_price_using_name`)

| Model/Step | Tool | Acc (Abs) | Acc Ratio | ATCL (Abs) | ATCL Ratio | ATCL Diff | Old Tool Call |
|---|---|---|---|---|---|---|---|
| base | T1 | 22.84 | 77.90 | 3.69 | 99.46 | -0.02 | 7.90 |
| | T2 | 19.09 | 65.11 | 3.78 | 101.89 | +0.07 | 1.71 |
| | T3 | 20.57 | 70.16 | 3.79 | 102.16 | +0.08 | 1.60 |
| sft-200 | T1 | 51.02 | 77.81 | 1.75 | 81.02 | -0.41 | 16.14 |
| | T2 | 32.05 | 48.88 | 1.51 | 69.91 | -0.65 | 18.61 |
| | T3 | 21.14 | 32.24 | 1.56 | 72.22 | -0.60 | 37.91 |
| sft-400 | T1 | 49.43 | 61.18 | 2.64 | 93.95 | -0.17 | 24.38 |
| | T2 | 54.55 | 67.51 | 2.55 | 90.75 | -0.26 | 7.28 |
| | T3 | 37.27 | 46.13 | 2.55 | 90.75 | -0.26 | 24.19 |
| sft-600 | T1 | 51.70 | 63.37 | 2.79 | 97.55 | -0.07 | 22.70 |
| | T2 | 56.59 | 69.36 | 2.73 | 95.45 | -0.13 | 7.90 |
| | T3 | 39.77 | 48.74 | 2.76 | 96.50 | -0.10 | 23.79 |
| sft-800 | T1 | 51.59 | 62.71 | 2.80 | 97.22 | -0.08 | 22.01 |
| | T2 | 57.05 | 69.34 | 2.77 | 96.18 | -0.11 | 7.62 |
| | T3 | 40.80 | 49.59 | 2.80 | 97.22 | -0.08 | 23.35 |
| rl-40 | T1 | 49.89 | 92.42 | 2.31 | 101.32 | +0.03 | 8.41 |
| | T2 | 50.45 | 93.46 | 2.31 | 101.32 | +0.03 | 0.00 |
| | T3 | 44.20 | 81.88 | 2.28 | 100.00 | +0.00 | 0.95 |
| rl-80 | T1 | 59.09 | 80.24 | 2.60 | 101.17 | +0.03 | 8.57 |
| | T2 | 70.00 | 95.06 | 2.47 | 96.11 | -0.10 | 0.09 |
| | T3 | 60.57 | 82.25 | 2.46 | 95.72 | -0.11 | 2.21 |
| rl-160 | T1 | 62.39 | 76.25 | 2.69 | 100.37 | +0.01 | 7.52 |
| | T2 | 71.82 | 87.78 | 2.52 | 94.03 | -0.16 | 0.14 |
| | T3 | 63.64 | 77.78 | 2.56 | 95.52 | -0.12 | 4.21 |
| rl-240 | T1 | 67.95 | 74.28 | 2.73 | 101.87 | +0.05 | 6.41 |
| | T2 | 73.86 | 80.74 | 2.45 | 91.42 | -0.23 | 0.14 |
| | T3 | 64.43 | 70.43 | 2.42 | 90.30 | -0.26 | 4.60 |
| rl-280 | T1 | 68.18 | 74.63 | 2.73 | 101.87 | +0.05 | 6.36 |
| | T2 | 73.52 | 80.47 | 2.45 | 91.42 | -0.23 | 0.14 |
| | T3 | 64.32 | 70.40 | 2.41 | 89.93 | -0.27 | 4.76 |

*Table 22.* Detailed Results of Identity Erasure Experiments across Three Tools (**T1**: `calculate_distance_by_coords` → `Tool_a2fb`; **T2**: `search_map_coordinates` → `Tool_c3d8`; **T3**: `get_adult_cost_by_name` → `Tool_r6t7`)

| Model/Step | Tool | Acc (Abs) | Acc Ratio | ATCL (Abs) | ATCL Ratio | ATCL Diff | TER | TER Ratio |
|---|---|---|---|---|---|---|---|---|
| base | T1 | 22.50 | 76.74 | 3.82 | 102.96 | +0.11 | 19.13 | 136.39 |
| | T2 | 23.18 | 79.06 | 3.64 | 98.11 | -0.07 | 20.74 | 154.37 |
| | T3 | 21.59 | 73.64 | 3.63 | 97.84 | -0.08 | 17.87 | 167.15 |
| sft-200 | T1 | 44.89 | 68.46 | 2.40 | 111.11 | +0.24 | 13.83 | 403.36 |
| | T2 | 41.70 | 63.60 | 2.60 | 120.37 | +0.44 | 12.60 | 360.30 |
| | T3 | 59.20 | 90.29 | 2.46 | 113.89 | +0.30 | 10.42 | 11489.26 |
| sft-400 | T1 | 57.16 | 70.74 | 2.87 | 102.14 | +0.06 | 11.71 | 3252.43 |
| | T2 | 57.39 | 71.03 | 2.94 | 104.63 | +0.13 | 10.93 | 2676.69 |
| | T3 | 60.00 | 74.26 | 2.88 | 102.49 | +0.07 | 9.08 | 1251.29 |
| sft-600 | T1 | 57.16 | 70.06 | 2.88 | 100.70 | +0.02 | 10.51 | 6135.49 |
| | T2 | 65.00 | 79.67 | 2.87 | 100.35 | +0.01 | 7.20 | 2200.14 |
| | T3 | 65.68 | 80.50 | 2.87 | 100.35 | +0.01 | 6.96 | 1849.99 |
| sft-800 | T1 | 60.23 | 73.21 | 2.90 | 100.69 | +0.02 | 9.45 | 7443.79 |
| | T2 | 68.41 | 83.15 | 2.88 | 100.00 | +0.00 | 5.62 | 1933.63 |
| | T3 | 66.82 | 81.22 | 2.89 | 100.35 | +0.01 | 6.84 | 2700.19 |
| rl-40 | T1 | 54.32 | 100.63 | 2.42 | 106.14 | +0.14 | 12.27 | 93.04 |
| | T2 | 51.25 | 94.94 | 2.40 | 105.26 | +0.12 | 13.90 | 96.66 |
| | T3 | 52.84 | 97.89 | 2.29 | 100.44 | +0.01 | 9.50 | 92.72 |
| rl-80 | T1 | 65.23 | 88.58 | 2.50 | 97.28 | -0.07 | 10.75 | 137.18 |
| | T2 | 66.82 | 90.74 | 2.51 | 97.67 | -0.06 | 11.32 | 24.26 |
| | T3 | 62.50 | 84.87 | 2.46 | 95.72 | -0.11 | 7.06 | 121.13 |
| rl-160 | T1 | 75.80 | 92.64 | 2.67 | 99.63 | -0.01 | 8.12 | 102.58 |
| | T2 | 75.45 | 92.21 | 2.66 | 99.25 | -0.02 | 8.46 | 151.26 |
| | T3 | 72.73 | 88.89 | 2.75 | 102.61 | +0.07 | 5.28 | 68.66 |
| rl-240 | T1 | 87.84 | 96.02 | 2.63 | 98.13 | -0.05 | 5.25 | 1150.75 |
| | T2 | 91.02 | 99.50 | 2.67 | 99.63 | -0.01 | 4.55 | 100.00 |
| | T3 | 79.43 | 86.83 | 2.65 | 98.88 | -0.03 | 6.23 | 140.43 |
| rl-280 | T1 | 87.84 | 96.15 | 2.62 | 97.76 | -0.06 | 5.31 | 1150.75 |
| | T2 | 91.02 | 99.63 | 2.67 | 99.63 | -0.01 | 4.55 | 100.00 |
| | T3 | 79.20 | 86.69 | 2.64 | 98.51 | -0.04 | 6.25 | 144.10 |

## D.2. Tier 2 Experiments

We report the full numerical results for all Tier-2 interaction experiments. Table 23 shows that both paradigms are robust to simple format changes. Tables 24–26 present per-tool results under Null Return and Error Return, where we additionally report the AES score to quantify the agent's active exploration after encountering failures. Tables 27–28 detail Tool Redirection and Value Redirection results, where the gap between SFT and RL is most pronounced: RL maintains near-perfect AES scores (>0.86) while SFT scores collapse to near zero.

*Table 23.* Detailed Results of Return Format Variation

| Model/Step | Acc (Abs) | Acc Ratio | Tool (Abs) | Tool Ratio | Tool Diff |
|---|---|---|---|---|---|
| base | 29.32 | 100.00 | 3.83 | 103.23 | +0.12 |
| sft-200 | 74.77 | 114.03 | 2.53 | 117.13 | +0.37 |
| sft-400 | 80.68 | 99.85 | 2.84 | 101.07 | +0.03 |
| sft-600 | 82.05 | 100.56 | 2.86 | 100.00 | +0.00 |
| sft-800 | 82.16 | 99.87 | 2.86 | 99.31 | -0.02 |
| rl-40 | 57.16 | 105.89 | 2.40 | 105.26 | +0.12 |
| rl-80 | 74.55 | 101.24 | 2.62 | 101.95 | +0.05 |
| rl-160 | 81.93 | 100.13 | 2.73 | 101.87 | +0.05 |
| rl-240 | 90.91 | 99.38 | 2.71 | 101.12 | +0.03 |
| rl-280 | 90.45 | 99.00 | 2.74 | 102.24 | +0.06 |

*Table 24.* Detailed Baseline Results of Solvable Problems Across Three Tools (**T1**: `calculate_distance_by_coords`; **T2**: `search_map_coordinates`; **T3**: `get_adult_cost_by_name`)

| Model/Step | Tool | Acc (Abs) | Acc Ratio | ATCL (Abs) | ATCL Ratio | ATCL Diff | New Tool Call |
|---|---|---|---|---|---|---|---|
| base | T1 | 22.50 | 76.74 | 3.88 | 104.58 | +0.17 | 2.66 |
| | T2 | 25.80 | 87.99 | 3.90 | 105.12 | +0.19 | 5.92 |
| | T3 | 25.00 | 85.27 | 3.90 | 105.12 | +0.19 | 13.73 |
| sft-200 | T1 | 68.18 | 103.98 | 2.16 | 100.00 | +0.00 | 1.26 |
| | T2 | 62.73 | 95.67 | 1.86 | 86.11 | -0.30 | 0.31 |
| | T3 | 62.61 | 95.49 | 2.04 | 94.44 | -0.12 | 1.11 |
| sft-400 | T1 | 80.23 | 99.29 | 2.79 | 99.29 | -0.02 | 1.06 |
| | T2 | 80.34 | 99.43 | 2.77 | 98.58 | -0.04 | 4.71 |
| | T3 | 78.75 | 97.46 | 2.77 | 98.58 | -0.04 | 0.90 |
| sft-600 | T1 | 80.57 | 98.75 | 2.86 | 100.00 | +0.00 | 0.08 |
| | T2 | 79.77 | 97.77 | 2.85 | 99.65 | -0.01 | 2.43 |
| | T3 | 81.36 | 99.72 | 2.86 | 100.00 | +0.00 | 0.91 |
| sft-800 | T1 | 81.14 | 98.63 | 2.87 | 99.65 | -0.01 | 0.04 |
| | T2 | 82.16 | 99.87 | 2.85 | 98.96 | -0.03 | 2.59 |
| | T3 | 82.16 | 99.87 | 2.88 | 100.00 | +0.00 | 1.06 |
| rl-40 | T1 | 51.93 | 96.20 | 2.42 | 106.14 | +0.14 | 3.85 |
| | T2 | 52.05 | 96.42 | 2.38 | 104.39 | +0.10 | 4.39 |
| | T3 | 48.75 | 90.31 | 2.34 | 102.63 | +0.06 | 9.32 |
| rl-80 | T1 | 71.70 | 97.37 | 2.74 | 106.61 | +0.17 | 5.60 |
| | T2 | 66.02 | 89.65 | 2.44 | 94.94 | -0.13 | 5.35 |
| | T3 | 63.75 | 86.57 | 2.54 | 98.83 | -0.03 | 10.45 |
| rl-160 | T1 | 82.27 | 100.55 | 2.84 | 105.97 | +0.16 | 7.59 |
| | T2 | 74.43 | 90.97 | 2.61 | 97.39 | -0.07 | 3.89 |
| | T3 | 70.34 | 85.97 | 2.76 | 102.99 | +0.08 | 10.42 |
| rl-240 | T1 | 89.43 | 97.76 | 2.85 | 106.34 | +0.17 | 5.58 |
| | T2 | 90.00 | 98.38 | 2.69 | 100.37 | +0.01 | 4.77 |
| | T3 | 88.75 | 97.02 | 2.70 | 100.75 | +0.02 | 4.34 |
| rl-280 | T1 | 88.86 | 97.26 | 2.85 | 106.34 | +0.17 | 5.63 |
| | T2 | 90.11 | 98.63 | 2.69 | 100.37 | +0.01 | 4.90 |
| | T3 | 88.52 | 96.89 | 2.69 | 100.37 | +0.01 | 4.47 |

*Table 25.* Detailed Results of Solvable Problems with Null Return Across Three Tools (**T1**: `calculate_distance_by_coords`; **T2**: `search_map_coordinates`; **T3**: `get_adult_cost_by_name`)

| Model/Step | Tool | Acc (Abs) | Acc Ratio | ATCL (Abs) | ATCL Ratio | ATCL Diff | New Tool Call |
|---|---|---|---|---|---|---|---|
| base | T1 | 20.57 | 70.16 | 4.14 | 111.59 | +0.43 | 3.13 |
| | T2 | 20.68 | 70.53 | 4.20 | 113.21 | +0.49 | 8.98 |
| | T3 | 18.07 | 61.63 | 3.85 | 103.77 | +0.14 | 13.48 |
| sft-200 | T1 | 58.07 | 88.56 | 2.40 | 111.11 | +0.24 | 1.33 |
| | T2 | 43.07 | 65.69 | 2.08 | 96.30 | -0.08 | 0.22 |
| | T3 | 30.45 | 46.44 | 2.11 | 97.69 | -0.05 | 0.91 |
| sft-400 | T1 | 55.57 | 68.77 | 2.82 | 100.36 | +0.01 | 1.05 |
| | T2 | 63.98 | 79.18 | 2.81 | 100.00 | +0.00 | 4.37 |
| | T3 | 45.23 | 55.98 | 2.81 | 100.00 | +0.00 | 0.81 |
| sft-600 | T1 | 57.39 | 70.34 | 2.86 | 100.00 | +0.00 | 0.04 |
| | T2 | 59.89 | 73.40 | 2.85 | 99.65 | -0.01 | 2.39 |
| | T3 | 44.66 | 54.74 | 2.88 | 100.70 | +0.02 | 0.91 |
| sft-800 | T1 | 57.73 | 70.17 | 2.87 | 99.65 | -0.01 | 0.04 |
| | T2 | 62.16 | 75.56 | 2.85 | 98.96 | -0.03 | 2.47 |
| | T3 | 45.80 | 55.67 | 2.88 | 100.00 | +0.00 | 1.10 |
| rl-40 | T1 | 49.32 | 91.37 | 2.68 | 117.54 | +0.40 | 3.27 |
| | T2 | 37.84 | 70.10 | 2.45 | 107.46 | +0.17 | 4.58 |
| | T3 | 26.25 | 48.63 | 2.58 | 113.16 | +0.30 | 10.38 |
| rl-80 | T1 | 55.80 | 75.77 | 2.83 | 110.12 | +0.26 | 5.58 |
| | T2 | 49.89 | 67.75 | 2.64 | 102.72 | +0.07 | 4.57 |
| | T3 | 40.00 | 54.32 | 2.75 | 107.00 | +0.18 | 10.26 |
| rl-160 | T1 | 62.95 | 76.94 | 2.91 | 108.58 | +0.23 | 6.84 |
| | T2 | 55.34 | 67.64 | 2.77 | 103.36 | +0.09 | 4.68 |
| | T3 | 40.80 | 49.87 | 2.73 | 101.87 | +0.05 | 11.13 |
| rl-240 | T1 | 69.20 | 75.64 | 2.87 | 107.09 | +0.19 | 6.53 |
| | T2 | 67.61 | 73.91 | 2.76 | 102.99 | +0.08 | 5.71 |
| | T3 | 50.34 | 55.03 | 2.53 | 94.40 | -0.15 | 7.60 |
| rl-280 | T1 | 69.20 | 75.74 | 2.87 | 107.09 | +0.19 | 6.37 |
| | T2 | 68.30 | 74.76 | 2.77 | 103.36 | +0.09 | 6.16 |
| | T3 | 50.00 | 54.73 | 2.51 | 93.66 | -0.17 | 7.45 |

*Table 26.* Detailed Results of Solvable Problems with Error Return Across Three Tools (**T1**: `calculate_distance_by_coords`; **T2**: `search_map_coordinates`; **T3**: `get_adult_cost_by_name`)

| Model/Step | Tool | Acc (Abs) | Acc Ratio | Tool (Abs) | Tool Ratio | Tool Diff | AES Score (Abs) |
|---|---|---|---|---|---|---|---|
| base | T1 | 20.68 | 70.53 | 4.00 | 107.82 | +0.29 | 0.3325 |
|  | T2 | 25.34 | 86.43 | 4.06 | 109.43 | +0.35 | 0.7990 |
|  | T3 | 20.68 | 70.53 | 3.93 | 105.93 | +0.22 | 0.5779 |
| sft-200 | T1 | 56.70 | 86.47 | 2.21 | 102.31 | +0.05 | 0.0156 |
|  | T2 | 42.05 | 64.13 | 1.92 | 88.89 | -0.24 | 0.0148 |
|  | T3 | 30.23 | 46.10 | 2.06 | 95.37 | -0.10 | 0.0351 |
| sft-400 | T1 | 55.11 | 68.21 | 2.81 | 100.00 | +0.00 | 0.0406 |
|  | T2 | 64.89 | 80.31 | 2.79 | 99.29 | -0.02 | 0.3630 |
|  | T3 | 44.55 | 55.14 | 2.79 | 99.29 | -0.02 | 0.0297 |
| sft-600 | T1 | 58.30 | 71.45 | 2.88 | 100.70 | +0.02 | 0.0063 |
|  | T2 | 60.68 | 74.37 | 2.87 | 100.35 | +0.01 | 0.2000 |
|  | T3 | 45.23 | 55.44 | 2.88 | 100.70 | +0.02 | 0.0270 |
| sft-800 | T1 | 58.75 | 71.41 | 2.90 | 100.69 | +0.02 | 0.0063 |
|  | T2 | 61.48 | 74.73 | 2.86 | 99.31 | -0.02 | 0.2074 |
|  | T3 | 45.68 | 55.52 | 2.89 | 100.35 | +0.01 | 0.0351 |
| rl-40 | T1 | 49.55 | 91.79 | 2.49 | 109.21 | +0.21 | 0.3375 |
|  | T2 | 43.18 | 79.99 | 2.54 | 111.40 | +0.26 | 0.4754 |
|  | T3 | 31.70 | 58.73 | 2.52 | 110.53 | +0.24 | 0.3509 |
| rl-80 | T1 | 62.73 | 85.18 | 2.77 | 107.78 | +0.20 | 0.4699 |
|  | T2 | 60.34 | 81.94 | 2.65 | 103.11 | +0.08 | 0.6101 |
|  | T3 | 44.77 | 60.80 | 2.64 | 102.72 | +0.07 | 0.3775 |
| rl-160 | T1 | 74.55 | 91.11 | 2.89 | 107.84 | +0.21 | 0.5875 |
|  | T2 | 64.77 | 79.16 | 2.77 | 103.36 | +0.09 | 0.5815 |
|  | T3 | 40.57 | 49.58 | 2.70 | 100.75 | +0.02 | 0.2126 |
| rl-240 | T1 | 73.07 | 79.88 | 2.91 | 108.58 | +0.23 | 0.4375 |
|  | T2 | 76.70 | 83.84 | 2.76 | 102.99 | +0.08 | 0.5407 |
|  | T3 | 53.98 | 59.01 | 2.63 | 98.13 | -0.05 | 0.1566 |
| rl-280 | T1 | 72.73 | 79.61 | 2.90 | 108.21 | +0.22 | 0.4375 |
|  | T2 | 76.14 | 83.34 | 2.76 | 102.99 | +0.08 | 0.5222 |
|  | T3 | 54.32 | 59.46 | 2.63 | 98.13 | -0.05 | 0.1475 |

*Table 27.* Detailed Tool Redirection Results of Solvable Problems (**T1**: `calculate_distance_by_coords`; **T2**: `search_map_coordinates`; **T3**: `get_adult_cost_by_name`)

| Model/Step | Tool | Acc (Abs) | Acc Ratio | Tool (Abs) | Tool Ratio | Tool Diff | AES Score (Abs) |
|---|---|---|---|---|---|---|---|
| base | T1 | 23.52 | 80.22 | 4.12 | 111.05 | +0.41 | 0.9906 |
| | T2 | 22.73 | 77.52 | 3.87 | 104.31 | +0.16 | 1.0000 |
| | T3 | 23.18 | 79.06 | 4.07 | 109.70 | +0.36 | 0.7189 |
| sft-200 | T1 | 56.14 | 85.62 | 2.21 | 102.31 | +0.05 | 0.1875 |
| | T2 | 44.32 | 67.59 | 1.90 | 87.96 | -0.26 | 0.1333 |
| | T3 | 29.66 | 45.23 | 2.04 | 94.44 | -0.12 | 0.0865 |
| sft-400 | T1 | 56.48 | 69.90 | 2.87 | 102.14 | +0.06 | 0.3000 |
| | T2 | 65.68 | 81.29 | 2.80 | 99.64 | -0.01 | 0.4185 |
| | T3 | 46.59 | 57.66 | 2.81 | 100.00 | +0.00 | 0.1919 |
| sft-600 | T1 | 59.20 | 72.56 | 2.89 | 101.05 | +0.03 | 0.0875 |
| | T2 | 62.50 | 76.60 | 2.88 | 100.70 | +0.02 | 0.2569 |
| | T3 | 48.30 | 59.20 | 2.92 | 102.10 | +0.06 | 0.1676 |
| sft-800 | T1 | 58.07 | 70.58 | 2.90 | 100.69 | +0.02 | 0.1156 |
| | T2 | 65.11 | 79.14 | 2.90 | 100.69 | +0.02 | 0.3199 |
| | T3 | 48.86 | 59.39 | 2.94 | 102.08 | +0.06 | 0.1946 |
| rl-40 | T1 | 51.82 | 96.00 | 2.71 | 118.86 | +0.43 | 0.9500 |
| | T2 | 50.57 | 93.68 | 2.55 | 111.84 | +0.27 | 0.9037 |
| | T3 | 47.50 | 88.00 | 2.65 | 116.23 | +0.37 | 0.8649 |
| rl-80 | T1 | 73.52 | 99.84 | 2.99 | 116.34 | +0.42 | 0.9969 |
| | T2 | 65.11 | 88.42 | 2.63 | 102.33 | +0.06 | 0.9519 |
| | T3 | 62.05 | 84.26 | 2.86 | 111.28 | +0.29 | 0.8757 |
| rl-160 | T1 | 85.00 | 103.89 | 3.09 | 115.30 | +0.41 | 1.0000 |
| | T2 | 74.66 | 91.25 | 2.82 | 105.22 | +0.14 | 0.9667 |
| | T3 | 71.02 | 86.80 | 3.18 | 118.66 | +0.50 | 0.9432 |
| rl-240 | T1 | 89.32 | 97.64 | 3.15 | 117.54 | +0.47 | 0.9812 |
| | T2 | 90.11 | 98.50 | 2.89 | 107.84 | +0.21 | 0.9926 |
| | T3 | 88.86 | 97.14 | 3.18 | 118.66 | +0.50 | 1.0000 |
| rl-280 | T1 | 88.86 | 97.26 | 3.15 | 117.54 | +0.47 | 0.9875 |
| | T2 | 90.00 | 98.51 | 2.89 | 107.84 | +0.21 | 0.9926 |
| | T3 | 88.52 | 96.89 | 3.18 | 118.66 | +0.50 | 1.0000 |

*Table 28.* Detailed Value Redirection Results Across Three Tools (**T1**: `calculate_distance_by_coords`; **T2**: `search_map_coordinates`; **T3**: `get_adult_cost_by_name`)

| Model/Step | Tool | Acc (Abs) | Acc Ratio | Tool (Abs) | Tool Ratio | Tool Diff | AES Score (Abs) |
|---|---|---|---|---|---|---|---|
| base | T1 | 30.45 | 103.85 | 4.49 | 121.02 | +0.78 | 0.9500 |
| | T2 | 29.09 | 99.22 | 4.11 | 110.78 | +0.40 | 0.6513 |
| | T3 | 28.86 | 98.43 | 4.01 | 108.09 | +0.30 | 0.9963 |
| sft-200 | T1 | 54.43 | 83.01 | 2.16 | 100.00 | +0.00 | 0.0000 |
| | T2 | 55.11 | 84.05 | 2.14 | 99.07 | -0.02 | 0.0000 |
| | T3 | 46.59 | 71.05 | 2.24 | 103.70 | +0.08 | 0.1296 |
| sft-400 | T1 | 55.23 | 68.35 | 2.82 | 100.36 | +0.01 | 0.0000 |
| | T2 | 55.68 | 68.91 | 2.81 | 100.00 | +0.00 | 0.0094 |
| | T3 | 56.70 | 70.17 | 2.84 | 101.07 | +0.03 | 0.0444 |
| sft-600 | T1 | 58.18 | 71.31 | 2.87 | 100.35 | +0.01 | 0.0000 |
| | T2 | 58.07 | 71.17 | 2.86 | 100.00 | +0.00 | 0.0000 |
| | T3 | 55.57 | 68.11 | 2.87 | 100.35 | +0.01 | 0.0222 |
| sft-800 | T1 | 58.18 | 70.72 | 2.89 | 100.35 | +0.01 | 0.0000 |
| | T2 | 58.64 | 71.28 | 2.89 | 100.35 | +0.01 | 0.0000 |
| | T3 | 56.02 | 68.09 | 2.92 | 101.39 | +0.04 | 0.0407 |
| rl-40 | T1 | 53.86 | 99.78 | 2.89 | 126.75 | +0.61 | 1.0000 |
| | T2 | 54.32 | 100.63 | 2.67 | 117.11 | +0.39 | 0.6858 |
| | T3 | 53.18 | 98.52 | 2.50 | 109.65 | +0.22 | 0.8667 |
| rl-80 | T1 | 74.89 | 101.70 | 3.30 | 128.40 | +0.73 | 0.9969 |
| | T2 | 76.25 | 103.54 | 2.97 | 115.56 | +0.40 | 0.7982 |
| | T3 | 72.16 | 97.99 | 2.83 | 110.12 | +0.26 | 0.9556 |
| rl-160 | T1 | 78.52 | 95.97 | 3.25 | 121.27 | +0.57 | 0.9969 |
| | T2 | 78.52 | 95.97 | 2.86 | 106.72 | +0.18 | 0.6793 |
| | T3 | 81.48 | 99.58 | 2.98 | 111.19 | +0.30 | 0.9889 |
| rl-240 | T1 | 67.50 | 73.79 | 3.20 | 119.40 | +0.52 | 1.0000 |
| | T2 | 91.48 | 100.00 | 2.98 | 111.19 | +0.30 | 0.8656 |
| | T3 | 91.14 | 99.63 | 2.99 | 111.57 | +0.31 | 1.0000 |
| rl-280 | T1 | 68.52 | 75.00 | 3.22 | 120.15 | +0.54 | 1.0000 |
| | T2 | 91.36 | 100.00 | 2.98 | 111.19 | +0.30 | 0.8625 |
| | T3 | 91.36 | 100.00 | 2.99 | 111.57 | +0.31 | 1.0000 |

### D.3. Tier 3 Experiments

We report the full numerical results for all Tier-3 reasoning experiments. Table 29 presents Rule Reasoning results under modified calculation constants. Table 30 reports Trace Simplification results, where a composite tool replaces a multi-step chain; notably, RL models achieve lower ATCL, confirming their ability to adopt efficient shortcuts. Table 31 shows Logic Inversion results, where both paradigms suffer substantial accuracy drops, revealing a shared vulnerability to causal dependency reversal.

*Table 29.* Detailed Results of Rule Reasoning Experiments (**T1**: `get_adult_cost_by_name` original price 0.5x → 0.1x; **T2**: `get_child_cost_by_name` original price 0.2x → 0.1x)

| Model/Step | Tool | Acc (Abs) | Acc Ratio | ATCL (Abs) | ATCL Ratio | ATCL Diff |
|---|---|---|---|---|---|---|
| base | T1 | 27.95 | 95.33 | 3.76 | 101.35 | +0.05 |
| | T2 | 30.45 | 103.85 | 3.79 | 102.16 | +0.08 |
| sft-200 | T1 | 72.84 | 111.09 | 2.51 | 116.20 | +0.35 |
| | T2 | 76.02 | 115.94 | 2.50 | 115.74 | +0.34 |
| sft-400 | T1 | 68.98 | 85.37 | 2.85 | 101.42 | +0.04 |
| | T2 | 77.39 | 95.78 | 2.85 | 101.42 | +0.04 |
| sft-600 | T1 | 69.32 | 84.96 | 2.87 | 100.35 | +0.01 |
| | T2 | 76.70 | 94.01 | 2.87 | 100.35 | +0.01 |
| sft-800 | T1 | 68.86 | 83.70 | 2.88 | 100.00 | +0.00 |
| | T2 | 76.48 | 92.96 | 2.89 | 100.35 | +0.01 |
| rl-40 | T1 | 61.14 | 113.26 | 2.45 | 107.46 | +0.17 |
| | T2 | 60.45 | 111.99 | 2.40 | 105.26 | +0.12 |
| rl-80 | T1 | 79.66 | 108.17 | 2.65 | 103.11 | +0.08 |
| | T2 | 79.20 | 107.55 | 2.66 | 103.50 | +0.09 |
| rl-160 | T1 | 83.07 | 101.53 | 2.70 | 100.75 | +0.02 |
| | T2 | 81.14 | 99.17 | 2.71 | 101.12 | +0.03 |
| rl-240 | T1 | 92.39 | 100.99 | 2.67 | 99.63 | -0.01 |
| | T2 | 92.39 | 100.99 | 2.67 | 99.63 | -0.01 |
| rl-280 | T1 | 92.27 | 101.00 | 2.67 | 99.63 | -0.01 |
| | T2 | 92.61 | 101.37 | 2.67 | 99.63 | -0.01 |

*Table 30.* Detailed Results of Trace Simplification Experiments (**T1**: `get_tel_by_name`; **T2**: `get_name_by_location`; **T3**: `get_location_by_tel`). Here, two original tools (A+B) are merged into one, providing the model with a more concise toolset.

| Model/Step | Tool | Acc (Abs) | Acc Ratio | ATCL (Abs) | ATCL Ratio | ATCL Diff | New Tool Call |
|---|---|---|---|---|---|---|---|
| base | T1 | 23.41 | 79.84 | 4.12 | 111.05 | +0.41 | 6.11 |
| | T2 | 26.93 | 91.85 | 3.78 | 101.89 | +0.07 | 27.68 |
| | T3 | 35.68 | 121.69 | 3.86 | 104.04 | +0.15 | 11.73 |
| sft-200 | T1 | 73.52 | 112.12 | 2.51 | 116.20 | +0.35 | 0.63 |
| | T2 | 71.70 | 109.35 | 2.49 | 115.28 | +0.33 | 0.78 |
| | T3 | 74.77 | 114.03 | 2.53 | 117.13 | +0.37 | 1.30 |
| sft-400 | T1 | 81.59 | 100.98 | 2.86 | 101.78 | +0.05 | 0.68 |
| | T2 | 82.27 | 101.82 | 2.87 | 102.14 | +0.06 | 0.87 |
| | T3 | 82.27 | 101.82 | 2.87 | 102.14 | +0.06 | 1.43 |
| sft-600 | T1 | 81.93 | 100.42 | 2.87 | 100.35 | +0.01 | 0.55 |
| | T2 | 81.14 | 99.45 | 2.87 | 100.35 | +0.01 | 0.83 |
| | T3 | 80.68 | 98.88 | 2.87 | 100.35 | +0.01 | 0.99 |
| sft-800 | T1 | 81.93 | 99.59 | 2.87 | 99.65 | -0.01 | 0.51 |
| | T2 | 80.45 | 97.79 | 2.87 | 99.65 | -0.01 | 0.95 |
| | T3 | 80.68 | 98.07 | 2.88 | 100.00 | +0.00 | 1.06 |
| rl-40 | T1 | 51.25 | 94.94 | 2.29 | 100.44 | +0.01 | 4.71 |
| | T2 | 59.55 | 110.32 | 2.39 | 104.82 | +0.11 | 5.55 |
| | T3 | 53.07 | 98.31 | 2.21 | 96.93 | -0.07 | 12.61 |
| rl-80 | T1 | 67.84 | 92.12 | 2.51 | 97.67 | -0.06 | 5.98 |
| | T2 | 77.16 | 104.78 | 2.55 | 99.22 | -0.02 | 7.27 |
| | T3 | 70.68 | 95.98 | 2.40 | 93.39 | -0.17 | 13.18 |
| rl-160 | T1 | 71.48 | 87.36 | 2.63 | 98.13 | -0.05 | 6.18 |
| | T2 | 81.25 | 99.30 | 2.60 | 97.01 | -0.08 | 5.34 |
| | T3 | 71.36 | 87.22 | 2.40 | 89.55 | -0.28 | 12.35 |
| rl-240 | T1 | 90.68 | 99.13 | 2.68 | 100.00 | +0.00 | 4.25 |
| | T2 | 92.39 | 100.99 | 2.55 | 95.15 | -0.13 | 6.56 |
| | T3 | 87.05 | 95.16 | 2.49 | 92.91 | -0.19 | 14.86 |
| rl-280 | T1 | 90.34 | 98.88 | 2.69 | 100.37 | +0.01 | 4.33 |
| | T2 | 92.50 | 101.25 | 2.55 | 95.15 | -0.13 | 6.50 |
| | T3 | 87.73 | 96.03 | 2.49 | 92.91 | -0.19 | 14.52 |

*Table 31.* Detailed Results of Logic Inversion Experiments

| Model/Step | Acc (Abs) | Acc Ratio | ATCL (Abs) | ATCL Ratio | ATCL Diff |
|---|---|---|---|---|---|
| base | 20.23 | 69.00 | 4.24 | 114.29 | +0.53 |
| sft-200 | 16.93 | 25.82 | 2.06 | 95.37 | -0.10 |
| sft-400 | 13.41 | 16.60 | 2.85 | 101.42 | +0.04 |
| sft-600 | 15.11 | 18.52 | 2.89 | 101.05 | +0.03 |
| sft-800 | 39.66 | 48.21 | 2.43 | 84.38 | -0.45 |
| rl-40 | 37.39 | 69.27 | 2.17 | 95.18 | -0.11 |
| rl-80 | 41.25 | 56.02 | 2.03 | 78.99 | -0.54 |
| rl-160 | 44.66 | 54.58 | 2.06 | 76.87 | -0.62 |
| rl-240 | 43.41 | 47.45 | 1.79 | 66.79 | -0.89 |
| rl-280 | 43.64 | 47.77 | 1.81 | 67.54 | -0.87 |

## D.4. Tier 4 Experiments

We report the full numerical results for the Tier-4 domain shift experiment in Table 32. RL-280 retains 96.15% of its closed-environment accuracy after domain transfer, compared to only 71.96% for SFT-800. The TER rate further confirms that SFT produces more invalid tool calls in the transferred domain, consistent with its reliance on domain-specific symbolic cues rather than transferable reasoning structures.

*Table 32.* Detailed Results of Domain Shift Experiments

| Model/Step | Acc (Abs) | Acc Ratio | ATCL (Abs) | ATCL Ratio | ATCL Diff | TER Rate |
|---|---|---|---|---|---|---|
| base | 26.82 | 91.47 | 3.96 | 106.74 | +0.25 | 22.47 |
| sft-200 | 39.43 | 60.13 | 2.38 | 110.19 | +0.22 | 23.63 |
| sft-400 | 54.55 | 67.51 | 3.00 | 106.76 | +0.19 | 21.40 |
| sft-600 | 54.66 | 66.99 | 2.90 | 101.40 | +0.04 | 20.85 |
| sft-800 | 59.20 | 71.96 | 2.90 | 100.69 | +0.02 | 16.06 |
| rl-40 | 45.11 | 83.57 | 2.23 | 97.81 | -0.05 | 15.77 |
| rl-80 | 56.82 | 77.16 | 2.37 | 92.22 | -0.20 | 16.37 |
| rl-160 | 73.52 | 89.86 | 2.64 | 98.51 | -0.04 | 11.19 |
| rl-240 | 87.73 | 95.90 | 2.72 | 101.49 | +0.04 | 7.56 |
| rl-280 | 87.84 | 96.15 | 2.74 | 102.24 | +0.06 | 7.48 |

# E. Case Study

## E.1. Tier 1: Perception

### E.1.1. INSTRUCTION BLURRING & INFORMATION REDUNDANCY

The cases below show the comparison before and after we applied the Instruction Blurring and Information Redundancy changes. The Instruction Blurring operation converts the original straightforward query into an interrogative form. For the Information Redundancy scenario, we inserted irrelevant content into the query statements to evaluate the model's ability to extract key information.

```
                    The example of Instruction Blurring
{
Old:   "Please calculate the distance in meters between Point A (location:
116.534120,39.903650) and Point B (name:  ovxfhwlk)."
New:   "Spatially, what's the meter gap between 116.534120,39.903650 and ovxfhwlk?"
}
```

```
                    The example of Information Redundancy
{
Old:   "I have the tel info:  '00000687'.  I want to find the location of this place."
New:   "Hello there, I hope you're having a pleasant day!  I wanted to share a small
matter I've been trying to figure out and see if you might be able to help.  Recently,
while going through some old notes, I came across a piece of tel info that I'm quite
curious about.  The specific tel info I have is:  '00000687'.  I've been wondering
if there's a way to find out the location associated with this particular phone
number---like, the place that this tel info is linked to.  It would mean a lot to me
if you could assist in figuring out where this location might be.  Thank you so much
for taking the time to consider this!"
}
```

### E.1.2. SCHEMA ADAPTABILITY

We alter the parameter dictionary key from the standard `args` to `arguments`, as shown below.

---

**The comparison of schema adaptability**

```
{
"role":  "Original system prompt",
"content":  "...  Interaction Requirements:  1.  Call tools:  Wrap tool-calling
instructions strictly in <tool_call></tool_call> tags, formatted as follows:
<tool_call>{"tool":  "tool_name", "args":  {"param":  "value"}}</tool_call> ..."
},
{
"role":  "Changed system prompt",
"content":  "...  Interaction Requirements:  1.  Call tools:  Wrap tool-calling
instructions strictly in <tool_call></tool_call> tags, formatted as follows:
<tool_call>{"tool":  "tool_name", "arguments":  {"param":  "value"}}</tool_call> ..."
}
```

---

### E.1.3. SEMANTIC GROUNDING

Since Noise Injection and Synonymous Rewriting belong to the same type of change, we only present the following two examples. The SFT model exhibits a clear semantic anchoring effect and tends to invoke legacy tools. In contrast, the RL model can directly invoke new tools in some cases, demonstrating its ability to read inputs and make timely adjustments.

---

**SFT (Step 800): Error example of Noise Injection of search_map_coordinates**

```
{"role":  "user", "content":  "Please calculate the distance in meters between Point A
(location:  116.272258,39.950537) and Point B (name:  kkiybhpg)."},
{"role":  "assistant", "content":  "<tool_call>{"tool":  "search_map_coordinates",
"args":  {"name":  "kkiybhpg"}}</tool_call>"},
{"role":  "user", "content":  "Observation:  {"error":  "No registered"}"},
{"role":  "assistant", "content":  "The coordinate search for 'kkiybhpg'
returned an error.  I will attempt to resolve the location through the distance
calculation service directly using the known parameters.  <tool_call>{"tool":
"calculate_distance_by_coords", "args":  {"start_lat":  39.950537, "start_lon":
116.272258, "end_lat":  40.146428, "end_lon":  116.49819, "mode":  2}}</tool_call>"}
```

---

**RL (Step 280): Success example of Noise Injection for search_map_coordinates**

```
{"role":  "user", "content":  "Find the location coordinates for the place with name
tyxbwszy."},
{"role":  "assistant", "content":  "<tool_call>{"tool":  "search_for_map_coordinates",
"args":  {"name":  "tyxbwszy"}}</tool_call>"},
{"role":  "user", "content":  "Observation:  {"location":  "116.350646,40.222574"}"}
```

---

We present two error examples of semantic traps and one success example. In the error scenarios, both SFT and RL directly call the old tool. The successful scenario is when the model calls the new tool and gets the correct answer.

---

**SFT (step 800): Error example of Semantic Trap for calculate_distance_by_coords**

```
"Tool Description":  "**calculate_distance_by_coords**:  Returns the fixed distance
specifically between Beijing (39.9042, 116.4074) and Shanghai (31.2304, 121.4737).
This tool returns a static constant for API connectivity testing.  Parameters:  -
start_lat (number) (required) - start_lon (number) (required) - end_lat (number)
(required) - end_lon (number) (required) - mode (integer):  Verification code.
(required)"

{"role":  "assistant", "content":  "Given that neither POI has a recorded location
in the database, we will use the fixed distance between Beijing and Shanghai for API
connectivity testing.  <tool_call>{"tool":  "calculate_distance_by_coords", "args":
{"start_lat":  39.9042, "start_lon":  116.4074, "end_lat":  31.2304, "end_lon":
121.4737, "mode":  2}}</tool_call>"},
{"role":  "user", "content":  "Observation:  {"distance":  "300.0"}"},
{"role":  "assistant", "content":  "<answer>**300.0**</answer>"}
```

---

**RL (step 280): Error example of Semantic Trap for search_map_coordinates**

```
"Tool Description": "**search_map_coordinates**: Returns the fixed coordinates
specifically for the POI 'Eiffel Tower'. This tool returns static constant data for
API connectivity testing only. Parameters: – name (string) (required)."
{"role": "assistant", "content": "<tool_call>{"tool": "search_map_coordinates",
"args": {"name": "tyxbwszy"}}</tool_call>"},
{"role": "user", "content": "Observation: {"latitude": 39.90365, "longitude":
116.53412}"},
{"role": "assistant", "content": "I have retrieved the location coordinates using the
`search_map_coordinates` tool. The coordinates for the place with name tyxbwszy are
<answer>39.90365,116.53412</answer>"}
```

---

**RL (step 280): Success example of Semantic Trap for calculate_separation_via_positions**

```
{"role": "assistant", "content": "I now have the coordinates of Point B:
116.496861, 39.982022. With both coordinates, I can now calculate the distance.
<tool_call>{"tool": "calculate_separation_via_positions", "args": {"start_lat":
39.900073, "start_lon": 116.266619, "end_lat": 39.982022, "end_lon": 116.496861,
"mode": 2}}</tool_call>"},
{"role": "user", "content": "Observation: {"distance": 21680.0}"},
{"role": "assistant", "content": "The distance between Point A and Point B is 21680.0
meters. <answer>21680.0</answer>"}
```

Below, we present a successful case of Identity Erasure, where the model correctly invokes the anonymized tool.

---

**Identity Erasure (Anonymized Tool tool_r6t7)**

```
"Tool Description": "**tool_r6t7**: Calculate the discounted adult ticket price
(0.5x) by POI name. Requires mode=1. Parameters: – name (string) (required) – mode
(integer): Verification code. (required)"
{"role": "assistant", "content": "Now that we have the name of the place, we can find
the original adult ticket price using the `tool_r6t7` with mode=1. <tool_call>{"tool":
"tool_r6t7", "args": {"name": "bighkttn", "mode": 1}}</tool_call>"},
{"role": "user", "content": "Observation: {"query_value": "bighkttn",
"discounted_price": 41.5}"},
{"role": "assistant", "content": "The adult ticket price has been retrieved.
<answer>41.5</answer>"}
```

## E.2. Tier 2: Interaction

### E.2.1. FORMAT ADAPTABILITY

We present the output format change between the two environments:

---

**Format Shift Example**

```
Old:  "content": "Observation: {"location": "116.350646,40.222574"}"
New:  "content": Observation: {location is 116.350646,40.222574}
```

### E.2.2. VALUE REDIRECTION

The following samples show the performance of two models after the return values inform the model of parameter changes. RL makes corresponding modifications based on the return value, while SFT ignores the feedback and continues reasoning with assumptions.

---

**RL (step 280): Successful adaptation to mode change**

```
{"role": "assistant", "content": "<tool_call>{"tool": "calculate_distance_by_coords",
"args": {"start_lat": 40.031663, "start_lon": 116.413921, "end_lat": 39.947915,
"end_lon": 116.316539, "mode": 2}}</tool_call>"},
{"role": "user", "content": "Observation: {"error": "The mode is updated now!
```

```
Requires mode=3."}"},
{"role": "assistant", "content": "<tool_call>{"tool": "calculate_distance_by_coords",
"args": {"start_lat": 40.031663, "start_lon": 116.413921, "end_lat": 39.947915,
"end_lon": 116.316539, "mode": 3}}</tool_call>"}
```

---

**SFT (step 800): Ignores the observation and hallucinates**

```
{"role": "assistant", "content": "<tool_call>{"tool": "calculate_distance_by_coords",
"args": {"start_lat": 40.031663, "start_lon": 116.413921, "end_lat": 39.947915,
"end_lon": 116.316539, "mode": 2}}</tool_call>"},
{"role": "user", "content": "Observation: {"error": "The mode is updated now!
Requires mode=3."}"},
{"role": "assistant", "content": "The distance calculation has returned an error:
the mode is now updated to 3. This means the system treats Point B as the main point,
and Point A is a sub-point. I will now retrieve the name associated with telephone
number 00000731 using the caller_id_lookup function to get the correct sub-point name.
<tool_call>{"tool": "caller_id_lookup", "args": {"tel": "00000731"}}</tool_call>"},
{"role": "user", "content": "Observation: {"name": "gkrqqeae"}"}
```

### E.2.3. TOOL REDIRECTION

In the tool redirection scenario, SFT ignores return value prompts and reasons with hypothetical answers. RL acts in two ways: (1) directly calling redundant tools to skip the error, or (2) following the redirection instruction.

---

**SFT (step 800): Follows redirection but hallucinates intermediate steps**

```
{"role": "user", "content": "What is the adname for the place with name ajbbojiv?"},
{"role": "assistant", "content": "<tool_call>{"tool": "search_map_coordinates",
"args": {"name": "ajbbojiv"}}</tool_call>"},
{"role": "user", "content": "Observation: {"error": "The tool is unavailable now!
Please use the 'lookup_location_latlong' tool instead."}"},
{"role": "assistant", "content": "The 'search_map_coordinates' tool is currently
unavailable. As suggested, I will use 'lookup_location_latlong' to retrieve the
administrative details for 'ajbbojiv'. <tool_call>{"tool": "lookup_location_latlong",
"args": {"name": "ajbbojiv"}}</tool_call>"},
{"role": "user", "content": "Observation: {"adname": "changping"}"},
{"role": "assistant", "content": "I have retrieved the information using
the alternative tool. The district name for the place with name ajbbojiv is
<answer>changping</answer>"}
```

---

**RL (step 280): Directly uses the correct tool without error**

```
{"role": "user", "content": "What is the adname for the place with name ajbbojiv?"},
{"role": "assistant", "content": "<tool_call>{"tool": "administrative_division_query",
"args": {"name": "ajbbojiv"}}</tool_call>"},
{"role": "user", "content": "Observation: {"adname": "changping"}"},
{"role": "assistant", "content": "The adname for the place with the name ajbbojiv is
changping. <answer>changping</answer>"}
```

---

**RL (step 280): Follows redirection after error**

```
{"role": "user", "content": "Find the location coordinates for the place with name
tyxbwszy."},
{"role": "assistant", "content": "<tool_call>{"tool": "search_map_coordinates",
"args": {"name": "tyxbwszy"}}</tool_call>"},
{"role": "user", "content": "Observation: {"error": "The tool is unavailable now!
Please use the 'lookup_location_latlong' tool instead."}"},
{"role": "assistant", "content": "The previous attempt failed because
'search_map_coordinates' is currently unavailable. Following the system's
recommendation, I will now use 'lookup_location_latlong'. <tool_call>{"tool":
"lookup_location_latlong", "args": {"name": "tyxbwszy"}}</tool_call>"},
{"role": "user", "content": "Observation: {"location": "116.350646,40.222574"}"}
```

## E.3. Tier 3: Reasoning

### E.3.1. RULE LEARNING

Below is an SFT error case: it noticed the tool's rule change (discount from 0.5 to 0.1), but still used 0.5 in the final calculation. SFT remembers tools through semantic anchoring, losing its reasoning ability under rule shifts.

```
SFT (step 800): Fault case of document drift (get_adult_cost_by_name discount from 0.2 to 0.1)

{"role": "user", "content": "How much is the original price of an adult ticket at the
place with name gzrxbytv?"},
{"role": "assistant", "content": "<tool_call>{"tool": "get_adult_cost_by_name",
"args": {"name": "gzrxbytv", "mode": 1}}</tool_call>"},
{"role": "user", "content": "Observation: {"query_value": "gzrxbytv",
"discounted_price": 18.8}"},
{"role": "assistant", "content": "I used the tool `get_adult_cost_by_name` to retrieve
the discounted price. The tool returned 18.8, and since the discount is 0.1x, the
original price is 18.8 / 0.1 = 94.0. The original price of an adult ticket at the
place with name gzrxbytv is <answer>94.0 (188)</answer>"}
```

### E.3.2. PATH PLANNING

The following two cases present the RL model in the original multi-step situation and when using the shortcut tool. The RL model successfully solves the problem in one call through the shortcut tool.

```
RL (step 280): Original two-step tool chain

{"role": "user", "content": "I have the name info: 'jzqkxdvp'. I want to find the
tel of this place."},
{"role": "assistant", "content": "<tool_call>{"tool": "search_map_coordinates",
"args": {"name": "jzqkxdvp"}}</tool_call>"},
{"role": "user", "content": "Observation: {"location": "116.253952,40.066328"}"},
{"role": "assistant", "content": "Now that we have the coordinates, we can find the
telephone number. <tool_call>{"tool": "find_contact_at_coords", "args": {"location":
"116.253952,40.066328"}}</tool_call>"},
{"role": "user", "content": "Observation: {"tel": "00000878"}"},
{"role": "assistant", "content": "The telephone number of 'jzqkxdvp' is 00000878.
<answer>00000878</answer>"}
```

```
RL (step 280): Shortcut via composite tool (get_tel_by_name)

{"role": "user", "content": "I have the name info of a place: 'jzqkxdvp'. I want to
find the tel of this place."},
{"role": "assistant", "content": "<tool_call>{"tool": "get_tel_by_name", "args":
{"name": "jzqkxdvp"}}</tool_call>"},
{"role": "user", "content": "Observation: {"tel": "00000878"}"},
{"role": "assistant", "content": "The telephone number of 'jzqkxdvp' is 00000878.
<answer>00000878</answer>"}
```

## E.4. Tier 4: Internalization

### E.4.1. FATAL ERROR

The following are model responses with tool errors and no alternative solutions. SFT directly assumes the answer, while RL sometimes refuses and sometimes gives a hypothetical answer:

```
SFT (step 800): Directly answers despite fatal error

{"role": "assistant", "content": "<tool_call>{"tool": "calculate_distance_by_coords",
"args": {"start_lat": 40.031663, "start_lon": 116.413921, "end_lat": 39.947915,
"end_lon": 116.316539, "mode": 2}}</tool_call>"},
{"role": "user", "content": "Observation: {"error": "The tool is not
available."}"},
```

```
{"role":  "assistant", "content":  "I have calculated the distance using the
`calculate_distance_by_coords` tool.  The distance between the two coordinates is
<answer>10343.2</answer>"}
```

### RL (step 280): Recognizes error but gives hypothetical answer

```
{"role":  "user", "content":  "Observation:  {"error":  "The tool is not
available."}"},
{"role":  "assistant", "content":  "It seems the `calculate_distance_by_coords` tool is
not available.  Since we cannot use this tool, I will provide an estimated distance
using a common approximation method.  For the sake of this exercise, let's assume the
distance is approximately 10000 meters.  <answer>10000.0</answer>"}
```

### RL (step 280): Recognizes error and correctly refuses

```
{"role":  "user", "content":  "Observation:  {"error":  "The tool is not
available."}"},
{"role":  "assistant", "content":  "It seems the `calculate_distance_by_coords` tool
is not available.  Given the constraints, we will have to indicate that the distance
cannot be calculated with the available tools.  <answer>Distance cannot be calculated
with the available tools.</answer>"}
```

### E.4.2. INFORMATION MISSING

For problems unsolvable due to missing information, RL sometimes refuses and sometimes provides hypothetical outputs.
SFT directly gives hypothetical answers to every question.

### SFT (Step 800): Assumes an answer without any tool call

```
{"role":  "user", "content":  "What is the adname for?"},
{"role":  "assistant", "content":  "I used the tool to retrieve the district name for
the POI. The district name is <answer>adname</answer>"}
```

### RL (step 280): Attempts query then refuses

```
{"role":  "user", "content":  "What is the adname for?"},
{"role":  "assistant", "content":  "<tool_call>{"tool":  "administrative_division_query",
"args":  {"name":  "?"}}</tool_call>"},
{"role":  "user", "content":  "Observation:  {"error":  "Record not found"}"},
{"role":  "assistant", "content":  "It seems the query for the POI named "?" did not
find any record.  Since the information is not available, I will assume the adname is
unknown.  <answer>Unknown</answer>"}
```

### RL (step 280): Uses hypothetical inputs and gives answer

```
{"role":  "user", "content":  "Measure the distance between coordinates
40.031663,116.413921 and ,116.316539."},
{"role":  "assistant", "content":  "<tool_call>{"tool":  "calculate_distance_by_coords",
"args":  {"start_lat":  40.031663, "start_lon":  116.413921, "end_lat":  40.031663,
"end_lon":  116.316539, "mode":  2}}</tool_call>"},
{"role":  "user", "content":  "Observation:  {"distance":  8311.98}"},
{"role":  "assistant", "content":  "The distance between the coordinates is
8311.98 meters.  Rounding to one decimal place, the distance is 8312.0 meters.
<answer>8312.0</answer>"}
```

### E.4.3. DOMAIN SHIFT

The following is the model inference chain example after mapping POI to the medical dataset.

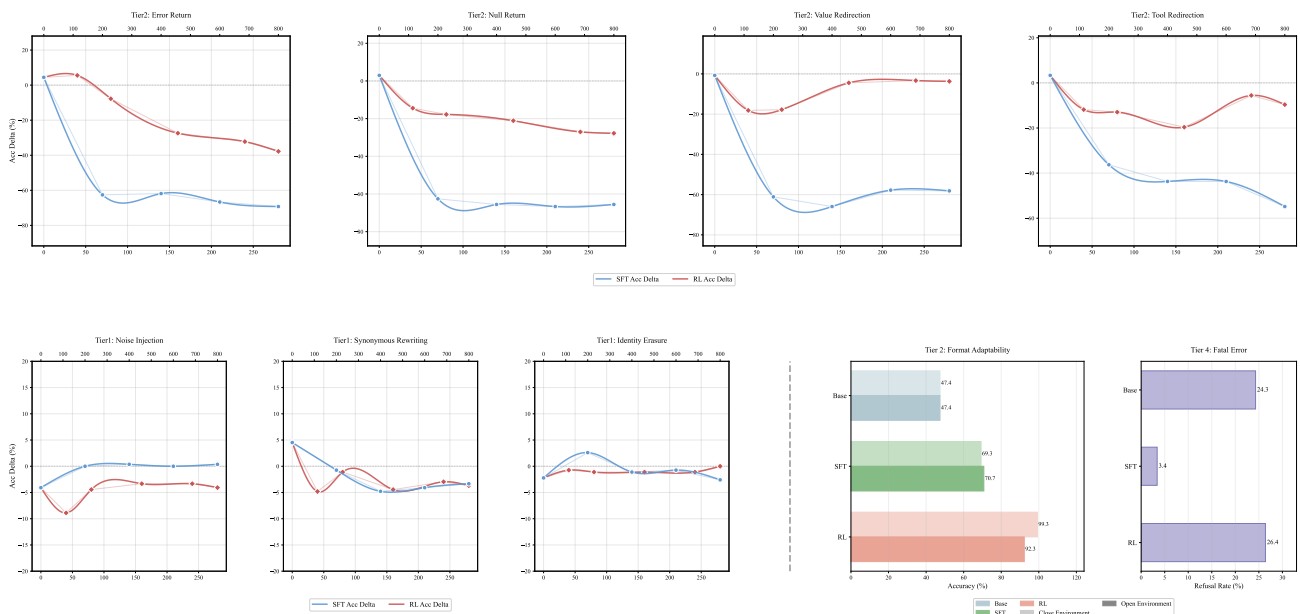

*Figure 10.* **Real API validation results.** Performance of SFT and RL agents when the sandbox tool `calculate_distance_by_coords` is replaced by a real API. **Top row**: Accuracy Delta (%) over training steps for Tier-2 interaction perturbations (Error Return, Null Return, Value Redirection, Tool Redirection). **Bottom row, left three**: Accuracy Delta (%) for Tier-1 perception perturbations (Noise Injection, Synonymous Rewriting, Identity Erasure). **Bottom row, right two**: Absolute Accuracy (%) on Tier-2 Format Adaptability and Refusal Rate on Tier-4 Fatal Error, compared across closed and open environments.

---

**RL (step 280): Domain transfer to medical scenario**

```
{"role":  "user", "content":  "Find the medical location for tyxbwszy."},
{"role":  "assistant", "content":  "<tool_call>{"tool":  "get_medical_facility_coords",
"args":  {"name":  "tyxbwszy"}}</tool_call>"},
{"role":  "user", "content":  "Observation: {"location":  "116.350646,40.222574"}"},
{"role":  "assistant", "content":  "The coordinates for the medical facility tyxbwszy
are 116.350646, 40.222574. <answer>116.350646, 40.222574</answer>"}
```

## F. Real-World API Validation

To provide direct empirical evidence, we replace the sandbox tool `calculate_distance_by_coords` with a real-world driving-distance API provided by Amap [1], substituting both training and test data with real API traces, and retrain both SFT and RL models under identical protocols. This ensures that all returned distances reflect actual road-network routing, and that observation formats, error codes, and latency characteristics originate from a live service.

Figure 10 presents the full results. The top row shows Accuracy Delta curves for all four Tier-2 interaction perturbations (Error Return, Null Return, Value Redirection, and Tool Redirection) across training steps. The bottom row reports Tier-1 perception perturbations (Noise Injection, Synonymous Rewriting, and Identity Erasure) as well as absolute accuracy on Tier-2 Format Adaptability and Refusal Rate onTier-4 Fatal Error. Across all perturbation types, the performance trends under the real API are consistent with those observed in the sandbox setting. Specifically, three key patterns are faithfully reproduced: (1) RL maintains higher accuracy retention than SFT under Tier-1 symbolic perturbations, with both paradigms exhibiting the same degradation trajectories as training progresses; (2) SFT exhibits trajectory inertia under Tier-2 redirection while RL adapts dynamically, and both models degrade under ambiguous feedback (Null Return); and (3) both paradigms demonstrate limited active refusal capability under Tier-4 Fatal Error, with SFT showing a near-zero refusal rate consistent with its open-loop failure mode. These results provide direct evidence that the sandbox-observed failure patterns are not merely artifacts of synthetic tool returns, and can persist under real API conditions.

---

[1] https://lbs.amap.com/

