# OpenReview forum: "Can Agents Generalize to the Open World? Unveiling the Fragility of Static Training in Tool Use"
_ICML.cc/2026/Conference — ICML 2026 regular_

### Official Review · Reviewer_DioS · 2026-02-13

**Soundness:** 3
**Presentation:** 2
**Significance:** 1
**Originality:** 1
**Overall Recommendation:** 3
**Confidence:** 4

**Summary:**

- The main motivatio is to see if LLM-based agents can effectively generalize to real, "dynamic" environemnts, as opposed to their "static" evaluation sandbox.
 - Specifically, the focus of the analysis is "tool-use" agents that undergo distributional shifts across query, action, observation, and domain dimensions.
 - Given their controlled environment, they do various analyses, including the ones on SFT and RL which suffer from various performance degradation.
 - This motivates their their approach of perturbation-augmented training.

**Compliance With Llm Reviewing Policy:**

Affirmed.

**Final Justification:**

The authors have addressed several of my concerns and hence I am increasing my score.
I continue to believe that this work would benefit from a better focus that being a hotchpotch of many small observations ("findings"?)

**Key Questions For Authors:**

See the previous box.

**Limitations:**

yes

**Strengths And Weaknesses:**

**Strength:** Overall, I agree that the issue of model robustness remains an open challenge and an important topic.

**Weakness** There are many works that study robustness to perturbations [1,2, among many others] -- Even in the context of RLHF/alignment [3, among many others]. Similarly, various modifications that you make, have been the subject of prior studies: instruction "blurring" (or under-specificity), redundancy, format adaptivity, etc. -- not necessarily in the context of tool-calling but broader range of challenges. And the conclusions is consistently that, all of these distributional modifications lead to performance degradation. Now why should one expect anything different in the context of "tool-use"??

Your purturbuaton-augmented training is also a natural corollary of the prior observations (again lots of related works on data-augmentation). Again I don't see any surprise here.

Overall, I don't see what new lesson(s) we can learn in this work.


[1] Gardner et al.  https://arxiv.org/abs/2004.02709
[2] Jia et al. https://arxiv.org/abs/1707.07328
[3] Shen et al. https://arxiv.org/abs/2309.16155

---

> ### Author Rebuttal · Authors · 2026-03-29
>
> Dear Reviewer,
>
> We respect your opinion, but believe this evaluation overlooks the fundamental differences between our work and the cited literature. We analyze the three papers you cited one by one:
>
> - **Gardner et al. [1]:** This work manually constructs contrastive sets on the test set for static NLP tasks (e.g., reading comprehension, sentiment analysis) to evaluate **the local decision boundaries of models**. Its core is the input perturbation of single-turn, passive classification/QA models.
> - **Jia et al. [2]:** This work inserts adversarial distracting sentences into SQuAD paragraphs to evaluate **the vulnerability of reading comprehension systems**. Similarly, it targets the text input perturbation of single-turn, passive QA models.
> - **Shen et al. [3]:** This work studies the consistency of reward models under semantically similar but different instructions, focusing on **the robustness of the reward model itself**, rather than the generalization ability of interactive agents.
>
> All three aforementioned works study **the behavior of passive, single-turn** **NLP** **models** **under input perturbations**. In contrast, our work studies **the behavior of interactive, multi-turn tool-use agents under distribution shifts**. There are essential differences between them:
>
> - **Differences in interactive dimensions:** Traditional robustness research only perturbs the input ($\Delta Q$), while our OpenAgent framework covers query shift ($\Delta Q$), action space shift ($\Delta A$), observation space shift ($\Delta O$), and cross-domain joint shift ($\Delta D$). Tool schema evolution, API deprecation, and environmental feedback changes are all unique challenges for tool-use agents.
> - **Differences in behavior patterns:** The failure of passive models is prediction error, while the failure modes of interactive agents are more complex—including "trajectory inertia," "boundary blindness," and hallucinatory forced completion. **These phenomena have not been discovered or discussed in existing robustness literature.**
> - **Novelty in comparing training paradigms:** We are the first to compare the differences in failure modes between SFT and RL in tool-use scenarios. We found mechanistic insights such as the "symbolic grounding" of SFT and the "teleological bias" of RL, **which have not been reported in any previous research.**
>
> As for PAFT, we acknowledge that data augmentation has a long history of application in ML. However, the core innovation of PAFT is not the concept of "augmentation" itself, but that **the three perturbation strategies are proposed based on the specific failure modes we diagnosed, with each strategy corresponding to addressing a specific layer of vulnerability**.
>
> Finally, we would like to reiterate our contributions:
>
> - We innovatively proposed the **OpenAgent setting**, which defines the types of changes that tool-calling Agents may encounter in open environments. Reviewers 2tB4, gqta, and sBe2 all believe that this setting is highly innovative and valuable for filling the current gap in the field.
> - We proposed a **four-layer evaluation framework of perception, interaction, reasoning, and internalization** capabilities to systematically diagnose and quantify the capability boundaries of agents in dynamic environments. Reviewers 2tB4, gqta, and sBe2 appreciated our framework and considered our experimental conclusions inspiring.
> - Finally, addressing the problem of poor performance of SFT in open settings, we proposed **PAFT, a lightweight and practical fine-tuning countermeasure**. Reviewers 2tB4 and gqta believe this method is simple and efficient and provides a good reference for subsequent work.
>
> We believe that **our work pioneers related research on the generalization of tool-use agents in open environments**. We hope the above explanation can help you more accurately understand the focus and innovative value of this paper, and we respectfully request that you give this paper a fair and objective evaluation.

---

> > ### Author Rebuttal · Reviewer_DioS · 2026-04-03
> >
> > I appreciate your detailed response. My view largely remains negative. Here I elaborate:
> >
> > On distinction from prior work:  Single-turn vs multi-turn: this is a fair point, but you need to articulate qualitative differences that differ from from the prior work. The same also goes with tool-use.  The fact that distribution cause performance degradation, and that augmentation-based training helps — is not new (whether you study it in a single turn or a billion-turn), UNLESS you show that the outcomes are different that what was understood from before. The authors have not demonstrated that the tool-use setting produces qualitatively distinct findings that could not have been predicted from prior robustness literature.
> >
> > The authors highlight terms like "trajectory inertia," "boundary blindness," and "symbolic anchoring" as novel discoveries. However, these are largely renamings of well-understood phenomena: For example, trajectory inertia: https://arxiv.org/abs/2501.17161 and several subsequent work. Boundary blindness is analogous to overconfidence under distribution shift (again, has its set of prior work ).
> >
> > The novelty, to my understand, here appears to be shallow. I just don't see how these "findings" add anything new to our collective understanding.
> >
> > PAFT: Training with data augmentation is, again, a standard practice. Augmentation strategies are routinely tailored to observed failure patterns. Many papers have exhausted this. Again, unclear what is the value-add.
> >
> > I maintain my current score, though I encourage the authors to more honestly situate their contributions.

---

> > > ### Author Response · Authors · 2026-04-03
> > >
> > > We sincerely appreciate your continued engagement and discussion. We address your remaining concerns point by point below:
> > >
> > > ### 1. Fundamental Differences in the Problem Setting
> > > You cited four works to suggest our research lacks a fundamental difference from existing robustness literature. However, the shared characteristic of those works is that each model inference is independent; there is no Markov chain of `action → environment → observation → next action`.
> > >
> > > We wish to reiterate the essential differences of our setting:
> > > * **Actions alter the environment state:** If an agent invokes an incorrect tool, the environment returns an error message or null value. The agent's subsequent decisions based on this faulty feedback will deviate further from the correct trajectory.
> > > * **Environmental feedback is an additional, dynamic shift source:** In traditional robustness research, input perturbations are predetermined. In agent scenarios, observation space changes (e.g., evolving API formats) are driven by the environment's evolution. The agent must parse these online and adjust its policy dynamically.
> > > * **Distinct error propagation patterns:** Errors in traditional models do not affect the next test sample. Agent errors propagate *within* a trajectory; a single incorrect tool call means all subsequent steps are built on a flawed premise.
> > >
> > > ### 2. Clarifications on Our Novel Discoveries
> > > You suggested concepts like *trajectory inertia* and *boundary blindness* are mere renamings of known phenomena. Due to space limits, we focus on these two to clarify the distinctions:
> > >
> > > * ***Trajectory inertia* is NOT the SFT memorization described in Chu et al. [4]:** They found SFT models suffer performance drops on unseen rule variants, which is a static observation of a capability boundary. Ours describes a dynamic phenomenon: during a multi-step trajectory, after receiving explicit error feedback, the agent continues executing subsequent steps according to its original, now-invalid plan. This behavior hinges entirely on "environmental feedback" and a "multi-step decision sequence." In single-turn QA tasks, the dilemma of adjusting subsequent steps after feedback does not exist. **Therefore, trajectory inertia cannot even be defined in traditional settings**.
> > > * ***Boundary blindness* is NOT a renaming of overconfidence:** Overconfidence refers to assigning high probability to an incorrect prediction, measurable in any single-step model. Our *boundary blindness* specifically describes an agent's behavior after encountering an unrecoverable fatal error: instead of recognizing the task is impossible, the agent loops retries or hallucinates a forced completion. **This involves two elements absent in traditional overconfidence**: (1) meta-cognition regarding tool-use capability boundaries, and (2) synthesizing multi-step interaction outcomes to deduce whether the task remains solvable.
> > >
> > > ### 3. Clarification on Research Value
> > > Your core stance appears to be: since distribution shift and data augmentation are studied in existing literature, applying similar concepts to multi-turn, environment-interacting agents lacks novelty.
> > >
> > > We respectfully argue this logic leads to a contradiction: **The NLP works you cited [1, 2] study robustness, had already been extensively studied in traditional machine learning classification tasks (as comprehensively summarized by Zhou [5]). If "re-studying a concept from field A in field B lacks value," applying traditional robustness concepts to NLP would similarly be devoid of contribution.**
> > >
> > > Yet, those NLP works are valued because the specific characteristics of language (e.g.semantic ambiguity) introduced new phenomena  unforeseen in traditional ML robustness research. Similarly, Chu et al. [4] extending the generalization to the post-training phase of foundation models is valuable because the unique properties of foundation models(e.g. scaling law) yield new discoveries.
> > >
> > > By the same logic, our work **extends robustness from passive NLP models to interactive tool-use agents**. This scenario introduces entirely new dimensions: **environment interaction, multi-step decision-making, state transitions, and dynamic action space evolution**. Many shift behaviors under these (like *trajectory inertia* and *Trace Simplification*) **do not and cannot exist in traditional settings**
> > > . We are not "renaming" concepts; we are uncovering new phenomena within a different computational paradigm.
> > >
> > > Once again we would like to thank you for careful suggestions, if you feel like our response resolve your concerns, please feel free to improve your rating.
> > >
> > > **Ref:**
> > >
> > > [4] Chu, T., Zhai, Y., Yang, J., Tong, S., Xie, S., Schuurmans, D., Le, Q. V., Levine, S., & Ma, Y.  SFT Memorizes, RL Generalizes: A Comparative Study of Foundation Model Post-training, 2025, In *Proceedings of the 42nd International Conference on Machine Learning (ICML)*.
> > >
> > > [5] Z.-H. Zhou. Open-environment machine learning. 2022,*National Science Review*,

---

### Official Review · Reviewer_sBe2 · 2026-02-16

**Soundness:** 3
**Presentation:** 2
**Significance:** 3
**Originality:** 3
**Overall Recommendation:** 5
**Confidence:** 4

**Summary:**

This paper presents an interesting empirical investigation of tool-use LLM agents under controlled environmental conditions. The authors construct a locally built Python sandbox environment featuring a predefined tool architecture, structured database, and dedicated tool-call modules for both RL training and evaluation. Experiments are conducted using Qwen2.5-7B-Instruct as the backbone model, with full-parameter SFT and GRPO-based RL as representative training paradigms.

The study systematically compares SFT-based and RL-based approaches and analyzes their respective failure behaviors under distribution shifts. A key contribution is the introduction of Perturbation-Augmented Fine-Tuning (PAFT), a disturbance-aware training strategy designed to improve robustness by injecting controlled perturbations during training.

The authors further organize their analysis into a four-tier diagnostic framework, enabling structured examination of agent failure modes across progressively deeper levels of capability. Overall, the paper provides a structured empirical study of robustness in tool-use agents and proposes a targeted training intervention to mitigate observed fragilities.

**Compliance With Llm Reviewing Policy:**

Affirmed.

**Final Justification:**

My concerns have been resolved. I have already given it an accept and my scores remain the same.

**Key Questions For Authors:**

The authors touch on a very new frontier on agent optimization. It is understandable that there are only a few relevant literatures. The authors assume the number of tools is static and predefined in the sandbox. Can the authors comment on how their work can be extended if dynamic tool discovery is allowed.

**Limitations:**

Yes

**Strengths And Weaknesses:**

They designed and implemented synthetic tool-use tasks in a controlled sandbox. The sample problem covers Geographical Point-of-Interest (POI) queries and simple calculations. I liked that they provided a very concrete example. This is their strong contribution.

Their discussions on 4 Tier Taxonomy Evaluation was confusing. Especially their terminology does not follow common usages in the lierature. For example their tier 1 is named Perception Generalization. “Perception” is metaphorical and strongly misleading (suggests sensory processing). It actually tests semantic interface interpretation. Tier 4: Internalization Generalization is also a problem. “Internalization” is abstract and non-standard in RL literature. It actually means boundary detection and epistemic awareness.

---

> ### Author Rebuttal · Authors · 2026-03-30
>
> Dear Reviewer,
>
> We appreciate your profound insight into the underlying technical essence. We are willing to continue the discussion with you below:
>
> - **On Tier 1 (Perception Generalization):** We respect your opinion, and you identify the technical essence as "**semantic interface interpretation**." While "Perception" is traditionally associated with visual or sensor data in robotics, for text-based LLM Agents, parsing **Queries, Tool Schemas, and Environment Observations** is their primary "sensory" mode of interaction. Thus, we use "Perception" to align with the **Perception-Reasoning-Action** cycle, emphasizing this as the initial stage where the agent ingests and decodes environmental signals.
> - **On Tier 4 (Internalization Generalization):** The terms "**boundary detection**" and "**epistemic awareness**" you provided are indeed precise technical descriptors. The reason we choose "internalization" is that it represents the peak of an agent's cognitive ability: it requires the agent to move beyond passive instruction-following and instead internalize its understanding of environmental rules and capability boundaries (World Model) into cognitive intuition (e.g., knowing when to refuse). This signifies an evolution from external rule-constraint to internal self-awareness.
>
> We hope we explain the reason for naming tiers, and we will carefully update the terminology and descriptions in the main text to achieve better clarity.
>
> **A(Question):** This is a highly insightful question. Based on our investigation, existing tool discovery methods [1,2,3] typically employ RAG or embedding-based retrieval to pre-filter tools for the model. Under such frameworks, the candidate toolset for a given turn is pre-determined. This essentially **reduces the problem to our experimental setting**: whether the provided candidate tools fall within the model's training distribution. Furthermore, in scenarios where the LLM must retrieve tools itself from extensive descriptions (similar to our "trajectory simplification" setup with all descriptions in the prompt), our empirical results demonstrate that **RL-based models exhibit stronger tool discovery and matching capabilities compared to SFT-based models.** In conclusion, as long as the distribution of training and testing is different, the conclusions of OpenAgent are of reference value. We hope the provided clarifications will sufficiently address your concerns.
>
> [1] Democratizing AI scientists using ToolUniverse https://arxiv.org/abs/2509.23426
>
> [2] AUTOTOOL: DYNAMIC TOOL SELECTION AND  INTEGRATION FOR AGENTIC REASONING https://arxiv.org/abs/2512.13278
>
> [3] TOOLMEM: Enhancing Multimodal Agents with Learnable Tool Capability Memory  https://arxiv.org/abs/2510.06664

---

> > ### Author Rebuttal · Reviewer_sBe2 · 2026-04-04
> >
> > The authros have fully addressed my concerns.

---

> > > ### Author Response · Authors · 2026-04-05
> > >
> > > Thank you very much for the positive assessment and your continued engagement. We deeply appreciate your acknowledgement that our clarifications on the taxonomy terminology and the discussion regarding dynamic tool discovery have fully addressed your concerns. We will carefully incorporate the refined terminology and the added analysis into the final version of the paper. Thank you again for your valuable feedback and contribution to improving our work.

---

### Official Review · Reviewer_gqta · 2026-03-12

**Soundness:** 3
**Presentation:** 2
**Significance:** 2
**Originality:** 2
**Overall Recommendation:** 4
**Confidence:** 3

**Summary:**

This paper addresses a critical problem in LLM agent development: agents trained on static benchmarks can be fragile when challenged with real-world distributional shifts. To diagnose the failure modes of LLM agents, the authors introduce an evaluation framework, OpenAgent, comprising four hierarchical tiers of generalization. Through well-designed experiments, the authors explore how SFT and RL models behave under various distributional shifts. Based on the discoveries, the authors propose PAFT, a data-centric intervention strategy that improves the robustness of LLM agents by injecting controlled perturbations into datasets during training.

**Compliance With Llm Reviewing Policy:**

Affirmed.

**Final Justification:**

The author's rebuttal addressed most of my concerns, so I will keep my scores unchanged.

**Key Questions For Authors:**

1. Can you provide some discussions about the sensitivity to some hyperparameters, such as the 30% perturbation rate in PAFT?
2. The experiments are conducted on Qwen2.5-7B-Instruct only. Can you explore how PAFT will work across different model families or scales?

**Limitations:**

yes

**Strengths And Weaknesses:**

Strengths:
1. This paper addresses a practical problem in training LLM agents, which lies in the gap between static benchmarks and dynamic, real-world environments.
2. This paper proposes a novel conceptual framework for evaluating the generalizability of LLM agents. The proposed OpenAgent framework formally characterizes distributional shifts across four dimensions, which is both novel and effective.
3. The proposed PAFT method offers a novel training methodology towards developing robust agents with a significant performance advantage.

Weaknesses:
1. The domain transfer experiment is limited to two domains. Experiments across more diverse domains would strengthen the claims.
2. The experiments focus on a single base architecture, and the generalizability of the insights across different model architectures is underexplored.
3. This paper lacks discussion about some critical practical considerations, including computational costs, training efficiency, and scalability to larger models.

---

> ### Author Rebuttal · Authors · 2026-03-31
>
> Dear Reviewer,
>
> We appreciate your valuable suggestion. Here is our reply to you:
>
> **W1:** To further evaluate the domain transfer capability of the model, we introduced two new domains (**Product** and **Factory**) while maintaining the underlying tool logic graph. We simultaneously modified the tool descriptions and user queries to increase complexity. As domain complexity increases, the base model’s performance remains relatively flat, while post-trained models exhibit varying degrees of decline. The results verified the conclusion that **RL model has a stronger ability to cope with open environments than the SFT model**.
>
> |Model|Baseline|Medical|Product|Factory|
> |:---|:---:|:---:|:---:|:---:|
> |Qwen2.5-7B|29.3|26.8|24.1|24.5|
> |SFT800|82.3|59.2|47.8|37.6|
> |RL280|91.4|87.8|76.5|71.8|
>
> W2: We would like to clarify that **a model's reasoning ability and its post-training paradigm** **are the key factors determining its generalization in open environments**. Compared with the scale (parameters) and version of the model, the model architecture (all of them are decoder-only architecture) has a relatively small impact on the reasoning ability (usually affecting the number of activation parameters and computational efficiency). To test the generalization of OpenAgent across models with different reasoning levels, we conducted experiments using **Qwen3-4B-Instruct** and **Qwen3-8B**. The results in the table below show a highly consistent pattern: **regardless of the model's reasoning ability, RL models show much stronger generalization than SFT models when facing open-world changes**. The Qwen3 base models already have strong **refusal capabilities**, showing that this is a focus of current pre-training. While RL training can retain some of this refusal ability, it tends to decline as training progresses. This is because the reward system focuses on finding solutions, which limits the model's tendency to refuse—consistent with our analysis in the paper. In summary, the findings of OpenAgent are **transferable across different model sizes** (reasoning levels) and provide a valuable reference for future research.
>
> **W3:** Regarding training efficiency and costs: under our experimental setup, SFT and RL training took approximately **3 hours** and **6 hours**, respectively, on an 8-node A800 GPU cluster. In all experiments, we trained the models until convergence to ensure an unbiased and fair comparison. We will include these detailed training statistics in the appendix of our revised manuscript.
>
> **Regarding Q1:** To address your question, we tested different PAFT data proportions (20% and 40%). **The results show that while more PAFT data improves generalization ability of models.** However, the benefits follow a pattern of diminishing returns. For instance, the improvement at 40% is similar to that at 30%.
>
> **Regarding Q2:** As we replied above, the reasoning ability affected by the number of model parameters and versions is the key to testing the generalization of conclusions, rather than the model architecture or model family. Thus, we conducted PAFT experiments on **Qwen3-4B-Instruct** and **Qwen3-8B** to address your concern. The results show that PAFT consistently improves the generalization of SFT models regardless of their size. This demonstrates that PAFT is transferable and works well even with models that have stronger reasoning capabilities.
>
> We hope these additional experiments address your concerns and we are happy to discuss this further.
>
> Table1: Ablation Study of PAFT. We evaluate the models' performance on four representive settings: Semantic Trap, Tool Redirection, Logic Inversion, and Fatal Error. To isolate the impact of specific components (e.g., EFP), we remove other settings while keeping the total data precent fixed at 30%.
>
> |Qwen2.5-7B(step)|Tier1(Acc Delta)|Tier2(Acc Delta)|Tier3(Acc Delta)|Tier4(Refusal Rate)|
> |:---|:---:|:---:|:---:|:---:|
> |SFT|-50.4|-45.3|-28.1|0.2|
> |PAFT(30%)|-4.12|-5.34|-9.82|99.6|
> |PAFT(20%)|-6.93|-7.73|-15.8|90.4|
> |PAFT(40%)|-5.34|-7.84|-11.1|96.4|
> |EFP|-30.1|-3.21|-27.0|1.7|
> |SBP|-38.9|-36.5|-31.7|99.2|
> |Tool Name|-14.9|-34.2|-10.6|2.12|
>
> Table 2: Analysis of Reasoning Capabilities across Model Scales. Due to the limitation of the response length, we reports three models per training paradigm.
>
> |Model(step)|Tier1(Acc Delta)|Tier2(Acc Delta)|Tier3(Acc Delta)|Tier4(Refusal Rate)|
> |:---|:---:|:---:|:---:|:---:|
> |**Qwen3-4B-Instruct**|-9.09|-7.04|7.01|62.3|
> |SFT200|-7.84|0.15|-14.59|31.9|
> |SFT600|-15.3|-8.40|-25.0|26.7|
> |SFT800|-18.7|-6.82|-24.8|22.7|
> |RL80|-5.14|1.91|-12.5|70.2|
> |RL160|-4.56|4.32|-12.5|82.9|
> |RL280|-5.91|2.72|-13.2|60.8|
> |PAFT800|-9.73|-6.93|-8.14|95.2|
> ||||||
> |**Qwen3-8B**|-2.96|-0.57|-3.41|64.6|
> |SFT200|-12.1|-5.80|-15.6|41.4|
> |SFT600|-14.4|-9.44|-22.8|20.2|
> |SFT800|-17.9|-13.8|-38.1|27.2|
> |RL80|-3.19|-1.82|0.45|56.9|
> |RL160|-4.09|-1.25|7.15|42.5|
> |RL280|-0.69|-0.79|-1.48|37.8|
> |PAFT800|-6.13|-3.56|-3.41|97.9|

---

> > ### Author Rebuttal · Reviewer_gqta · 2026-04-03
> >
> > Thanks for the authors, and I will keep my score.

---

> > > ### Author Response · Authors · 2026-04-03
> > >
> > > We sincerely thank you for your continued engagement and for taking the time to review our rebuttal.
> > >
> > > We noticed your ack selection of option (c), indicating that some of your concerns remain partially resolved or unresolved. As we highly value your feedback and strive to improve the quality of our work, **we would be deeply grateful if you could briefly specify which aspects of our paper or rebuttal still leave you with doubts**.
> > >
> > > We are more than happy to provide any further clarification, discussion, or additional evidence to fully address your remaining questions.
> > >
> > > Thank you again for your time, effort, and constructive evaluation.
> > >
> > > P.S. Please note that due to the constraints of the ICML discussion system, we may be limited in our ability to post new, separate replies in this thread. Should you have any further questions or follow-ups, we will actively address them by editing this current response. We will be monitoring the discussion closely.

---

### Official Review · Reviewer_2tB4 · 2026-03-12

**Soundness:** 3
**Presentation:** 3
**Significance:** 2
**Originality:** 2
**Overall Recommendation:** 4
**Confidence:** 3

**Summary:**

The paper studies whether current LLM agents trained with SFT or RL can generalize to open-world tool-use settings where queries, tools, and interactions shift. The authors formalize an OpenAgent setting and design a controlled sandbox that introduces structured perturbations across perception, interaction, reasoning, and internalization levels. Based on their analysis of failure modes, they propose Perturbation-Augmented Fine-Tuning, which injects controlled anomalies into training trajectories to improve robustness under distribution shifts.

**Compliance With Llm Reviewing Policy:**

Affirmed.

**Final Justification:**

This paper addresses an important and timely problem, and I find the OpenAgent framing, structured perturbation analysis, and controlled sandbox evaluation to be meaningful contributions. The rebuttal helped clarify perturbation sensitivity and strengthened the PAFT ablation discussion, but my main concerns remain only partially resolved: the link from the sandbox setting to real evolving APIs is still not directly validated, and some mechanism-level claims, especially around RL-specific failure causes, remain more suggestive than conclusive. Overall, the rebuttal reinforced rather than changed my view, and I maintain my original recommendation.

**Key Questions For Authors:**

1. I’m curious how well the OpenAgent evaluation correlates with real-world deployment. Have the authors tested these findings on real APIs or real tool ecosystems that evolve over time?
2. I wonder how sensitive the results are to the specific perturbation design. If the perturbations change slightly, do we observe similar failure patterns?
3. For PAFT, what kinds of perturbations help the most? For example, are observation noise, tool schema changes, or reasoning perturbations the most important?
4. Would PAFT still be effective for much larger agents or more complex multi-tool workflows?

**Limitations:**

yes

**Strengths And Weaknesses:**

Soundness

Positive:
1. I think the experimental setup is generally careful and systematic. The authors try to isolate different types of environmental shifts (query, tool, interaction), which makes the analysis more convincing than just showing aggregated performance drops.
2. I like the idea of building a controlled sandbox environment. It allows the authors to analyze failure modes in a more diagnostic way instead of relying on noisy real-world APIs.
3. The comparison between SFT and RL agents is quite informative. The paper does a good job highlighting that the two paradigms fail in different ways, which gives some mechanistic insight.

Negative:
1. I feel some claims about the causes of failures (like “teleological bias in RL reward structures”) are a bit speculative. The experiments show performance drops, but I’m not fully convinced the analysis fully proves these specific mechanisms.
2. The evaluation environment, although controlled, still feels synthetic. I’m not sure how well these perturbations correspond to real-world tool evolution or API changes.
3. The PAFT method seems relatively simple (injecting anomalies and symbolic noise), but the paper does not fully analyze why it works or when it might fail. A deeper ablation or theoretical explanation would make the claims stronger.

Presentation

Positive:
1. Overall I think the paper is clearly structured. The introduction sets up the problem well and motivates why static benchmarks are insufficient.
2. The idea of organizing shifts into a hierarchy (perception, interaction, reasoning, internalization) is quite helpful conceptually. It makes the evaluation framework easier to understand.

Negative:
1. Some sections feel a bit heavy on terminology. I sometimes felt the framework names and categories made the paper slightly harder to read than necessary.
2. The description of PAFT is a bit brief. I had to read the section carefully to fully understand how perturbations are injected and how they differ from standard data augmentation.

Significance

Positive:
1. I think the core question is very important: whether tool-using agents actually generalize beyond static training distributions. This is a big issue for real-world deployment.
2. The idea of explicitly modeling open-world shifts is valuable. Many benchmarks assume a fixed environment, so this work pushes the community to think about non-stationary settings.
3. The diagnostic framework could potentially become a useful evaluation tool for future agent research.

Negative:
1. The empirical gains of PAFT seem modest. It improves robustness, but it’s not clear whether it fundamentally solves the open-world generalization issue.
2. The work focuses mainly on tool-use agents. I’m not fully sure how broadly the findings transfer to other types of LLM agents or planning systems.
3. Some of the conclusions might depend heavily on the specific sandbox design, which may limit the generality of the results.

Originality

Positive:
1. I think formalizing the “OpenAgent” setting is a nice conceptual contribution. It frames the generalization problem in a clearer way.
2. The structured perturbation framework across multiple levels (perception, interaction, reasoning, internalization) is a creative way to analyze agent failures.
3. The idea of training with perturbation-injected trajectories is a practical and simple idea that could inspire follow-up work.

Negative:
1. The core method feels similar to robustness training or data augmentation ideas already common in ML. The novelty mainly lies in how it is applied to tool-use agents.
2. The perturbation strategy itself is not very technically sophisticated, so the methodological novelty is moderate.
3. Some of the conceptual framing overlaps with prior discussions of distribution shift and robustness in RL and agent systems, though the paper packages it nicely for LLM agents.

---

> ### Author Rebuttal · Authors · 2026-03-31
>
> Dear Reviewer,
>
> We sincerely appreciate your positive and constructive feedback. For better clarity, we have summarized your questions and provided our responses as follows:
>
> - **Consistency between Sandbox Tools and Real-world APIs（Soundness N2 & Significance N3 & Q1）**: For LLM Agents, **there is no fundamental difference between real-world APIs and synthetic sandbox tools**; both function as external interfaces to be invoked. By using sandbox tools, we simulate structured perturbations prevalent in reality, such as schema evolution and noise injection, **while circumventing the instability and uncontrollability of live APIs**. Notably, ToolBench [1] initially employed real APIs but eventually introduced Stable-ToolBench [2] using LLM as answer sandboxe due to API deprecation. Thus, the conclusion of OpenAgent can be extended to real-world scenarios, and the sandbox design will not limit its applicability.
> - **The generalization of OpenAgent in other type of Agents (Significance N2)**: While our study focuses on tool-use, this task serves as a representative paradigm for all LLM agents interacting with external environments. The fundamental challenge, adapting to **unseen instructions and dynamic environment feedback (OOD scenarios)**, is shared by other agents and complex planning systems alike. **Our findings hold important reference value for LLM-based agents that operate in open environments and suffer from the train and test distribution shift**. In future work, we added a discussion in the revised manuscript to further elaborate on how these insights can be applied to broader agentic and planning frameworks.
> - **The Influence of Perturbations of different degrees (Q2)**: Firstly, we defined quantitative metrics for perturbations at different levels (e.g., semantic similarity, effective character count, and the number of new tools). For example, we utilized Qwen-Embedding-v3 to measure vector distances, representing the degree of perturbation change. **When the types or degrees of perturbations vary slightly, we observe highly consistent failure patterns across different settings**. This alignment further validates the broad applicability of the OpenAgent framework.
>
> |Tier 1(Acc)|origin|68.22%|81.68%|96.19%|
> |:---|:---:|:---:|:---:|:---:|
> |base|29.32|29.77|27.88|26.74|
> |SFT800|77.27|74.66|72.05|77.84|
> |RL280|91.36|87.05|72.50|90.68|
>
> |Tier 2(Acc)|origin|92.4%|54.4%|23.6%|
> |:---|:---:|:---:|:---:|:---:|
> |base|29.32|29.31|26.53|21.36|
> |SFT800|82.27|82.16|75.34|76.59|
> |RL280|91.36|90.45|89.29|86.93|
>
> |Tier 3(Average Tool Chain Length)|origin|tool+1|tool+2|tool+3|
> |:---|:---:|:---:|:---:|:---:|
> |base|3.73|3.79|3.64|3.59|
> |SFT800|2.94|2.94|2.94|2.93|
> |RL280|2.74|2.66|2.58|2.33|
>
> - **The brief description of PAFT in the main text and its design motivation:** Due to space constraints in the main text, our primary focus was on problem defination and empirical analysis, and we will provide a comprehensive description of PAFT in the Appendix. The design motivation behind PAFT is to expand the model's potential response paths in real-world scenarios (e.g., simulating tool errors or refusals). To further address your concerns about importance of the part, we conducted an ablation study on the three data synthesis strategies: **EFP, SBP, and Tool Name** (please refer to Reviewer gqta for details). The results show that each strategy enhances the model's ability to handle different types of changes, e.g. SBP improves the model's ability to refuse invalid requests. Moreover, the three modules are equally important and can be adjusted according to downstream tasks. For Q4, when encountering **more complex tool invocation chains**, we only need to add corresponding enhancement strategies accordingly to enable PAFT to take effect. For Significance N1, PAFT provides a scalable training paradigm focused on "interaction logic" rather than simple data-fitting, offering a robust foundation for open-world generalization. PAFT may not cover all open-world scenarios, but its modular design provides an extensible foundation, and we will continue to expand it in future research.
> - **Soundness N1 & Presentation N1**: In the main text (Lines 407-414), we explicitly demonstrate how reward functions purely focused on goal attainment can inadvertently constrain the model’s epistemic awareness. In the revised manuscript, we will reduce the number of terminology.
>
> We hope these additional experiments address your concerns and we are happy to discuss this further.
>
> [1] Toolllm: Facilitating large language models to master 16000+ real-world APIs. ICLR 2024. https://arxiv.org/abs/2307.16789
>
> [2]  Stabletoolbench: Towards stable large-scale benchmarking on tool learning of large language models. Findings of ACL 2024. https://arxiv.org/abs/2403.07714

---

> > ### Author Rebuttal · Reviewer_2tB4 · 2026-04-02
> >
> > I thank the authors for the thoughtful rebuttal. The added clarification on perturbation sensitivity and the extra PAFT ablation help address my concerns about robustness analysis and presentation, but the core issues are still only partially resolved: the paper still does not provide direct evidence that the sandbox conclusions transfer to real evolving APIs, the claim about RL’s “teleological bias” remains more suggestive than fully demonstrated, and the discussion of larger agents or more complex workflows is still speculative. My main follow-up question is whether the authors can provide any concrete real-API validation, or at least a stronger empirical argument that the proposed perturbations faithfully track real deployment shifts. Given this, I do not change my original weak accept score.

---

> > > ### Author Response · Authors · 2026-04-05
> > >
> > > We sincerely appreciate your recognition of our work and your continued engagement!  We address your three remaining concerns below.
> > >
> > > ### I. Transfer of Sandbox Conclusions to Real-World APIs
> > > Using sandboxes for agent training and evaluation is standard practice. ToolBench [1] initially used 16,000+ real RapidAPIs but faced widespread deprecation, prompting StableToolBench [2] to simulate responses for reproducibility. MINT [3], GTA [4], and BFCL V3 [5] also adopt sandbox or local codebases. Our sandbox doesn't sidestep real-world scenarios; rather, **OpenAgent's core value is systematically revealing failure patterns in controlled environments to guide real-world deployment reliably**.
> > >
> > > Nonetheless, to provide direct empirical evidence, we replaced our tool of `calculate_distance_by_coords` from local computation to a real driving distance API [6], substituting both training and test data with real API traces, and retrained accordingly. The results on  perturbations related with tools are as follows:
> > >
> > > [Supplementary Material](https://hilarious-bublanina-aac784.netlify.app/rebuttal.pdf)
> > >
> > > The performance trends under real API perturbations are **highly consistent with the sandbox**, validating the generalizability of our conclusions.
> > >
> > > We also further **surveyed public changelogs from major API platforms** and mapped real breaking changes to our perturbation taxonomy:
> > >
> > > | **Real API Change**| **OpenAgent Shift**  |
> > > | ------------------------------------------------------------ | ----------------------------------------------- |
> > > | OpenAI renamed functions/function_call to tools/tool_choice  | Tier 1: Schema Adaptability  |
> > > | OpenAI deprecated /v1/fine-tunes, replaced with /v1/fine_tuning/jobs | Tier 1: Symbolic Drift |
> > > | Deprecated OpenAI model snapshots return errors  | Tier 2: Error Return|
> > > | OpenAI Assistants API v1→v2; v1 returns deprecation notice with migration guidance | Tier 2: Tool Redirection                        |
> > > | Google Maps API v2→v3; response format XML→JSON  | Tier 2: Format Shift|
> > > | Twitter/X API v1.1 endpoints deprecated, returning 410/403   | Tier 2: Error Return + Tier 4: Fatal Error |
> > >
> > > Much tier has concrete real-world counterparts, confirming that our perturbation design is a systematic abstraction of documented API change patterns.
> > >
> > > ### II. Empirical Evidence for RL's Teleological Bias
> > >
> > > This conclusion is drawn from our fine-grained analysis under Tier 4 Fatal Error scenarios. As shown in Appendix E.4.1, SFT and RL exhibit different failure mechanisms. Upon receiving an error, SFT directly outputs "...distance is \<answer\>..." without mentioning the error at all. It treats the error observation as a normal return value, representing a *perception-level* failure.
> > >
> > > In contrast, RL explicitly states "It seems the...tool is not available", indicating it clearly perceives the error. However, it immediately fabricates an answer rather than refusing to answer. This behavioral pattern is direct evidence of *teleological bias*. Because RL is optimized solely for task completion, its policy learns to "provide an answer at all costs": prioritizing completion over honest refusal, which is distinct from mere overconfidence.
> > >
> > > To quantify this, we sampled 100 non-refusal cases under Fatal Errors to measure the *error awareness* rate :
> > >
> > > |Metric|Baseline|SFT-800|RL-280|
> > > |---|---|---|---|
> > > |Error awareness rate|45%|3%|93%|
> > >
> > > This comparison further proves two qualitatively distinct failure mechanisms: **(1) SFT fails at the perception level (open-loop, failing to parse environmental feedback); (2) RL fails at the decision level (closed-loop but driven by teleological bias, prioritizing completion over refusal)**.
> > >
> > > ### III. Larger Models and More Complex Workflows
> > > Due to computational constraints, we cannot train larger models (e.g., 32B) within the rebuttal period. However, we have validated our conclusions across different architectures  and scales, as detailed in our response to Reviewer gqta (W1). **The identified failure patterns consistently reproduce, confirming the generalizability of our findings**.
> > >
> > >
> > > Once again we would like to thank you for careful suggestions, if you feel like our response resolve your concerns, please feel free to improve your rating, thank you very much in advance!.
> > >
> > > **Ref:**
> > >
> > > [1] Qin et al. ToolLLM: Facilitating Large Language Models to Master 16000+ Real-world APIs. ICLR 2024.
> > >
> > > [2] Guo et al. StableToolBench: Towards Stable Large-Scale Benchmarking on Tool Learning of Large Language Models. Findings of ACL 2024.
> > >
> > > [3] Wang et al. MINT: Evaluating LLMs in Multi-turn Interaction with Tools and Language Feedback. ICLR 2024.
> > >
> > > [4] Wang et al. GTA: A Benchmark for General Tool Agents. NeurIPS 2024.
> > >
> > > [5] Patil et al. The Berkeley Function Calling Leaderboard (BFCL): From Tool Use to Agentic Evaluation of Large Language Models. ICML 2025.
> > >
> > > [6] https://lbs.amap.com/

---

### Decision · Program_Chairs · 2026-04-30

**Decision:**

Accept (regular)

**Comment:**

This paper introduces OpenAgent, a diagnostic framework for agent generalization built around a four tier perturbation hierarchy and a controlled sandbox for injecting structured failures. It provides a systematic decomposition of SFT and RL failure modes, most notably showing a striking gap in error awareness under fatal perturbations (3% for SFT vs 93% for RL), and proposes PAFT, a targeted data augmentation strategy aligned with different failure tiers.

Reviewer scores were split but mildly positive overall (4/4/5/3). Strengths include the clean experimental setup, the reusable framework, and the SFT vs RL comparison. Concerns focused on limited novelty relative to prior robustness work, unclear terminology, restricted evaluation scope, and whether sandbox perturbations reflect real world conditions. The rebuttal significantly strengthened the paper with real API validation, cross scale experiments, domain transfer results, and a strong defense of the novelty as specific to interactive multi step agents. Positive reviewers maintained their scores, while the main dissent remained unconvinced on novelty.

Overall, I lean toward accept based on the reviewers' responses. The key contribution is the quantitative error-awareness decomposition, a concrete and novel result specific to interactive agents that is not captured in prior single-turn robustness work. The real API mapping further strengthens the practical relevance, and the four-tier framework appears reusable. That said, the paper would benefit from clearer framing: aligning claims with actual contributions, refining terminology, and foregrounding the strongest empirical findings.